# Structural analyses at pseudo atomic resolution of Chikungunya virus and antibodies show mechanisms of neutralization

Siyang Sun[1†‡a], Ye Xiang[1†‡b], Wataru Akahata[2], Heather Holdaway[1‡c], Pankaj Pal[3], Xinzheng Zhang[1], Michael S Diamond[3], Gary J Nabel[2], Michael G Rossmann[1]*

[1]Department of Biological Sciences, Purdue University, West Lafayette, United States; [2]Vaccine Research Center, National Institute of Allergy and Infectious Diseases, National Institutes of Health, Bethesda, United States; [3]Departments of Medicine, Molecular Microbiology, Pathology & Immunology, Washington University School of Medicine, St Louis, United States

*For correspondence: mr@purdue.edu

†These authors contributed equally to this work

‡Present address: [a]8211 Scicor Drive, Indianapolis, United States; [b]School of Medicine, Tsinghua University, Beijing, China; [c]Cleveland Center for Membrane and Structural Biology, Case Western Reserve University, Cleveland, United States

**Competing interests:** The authors declare that no competing interests exist.

**Abstract** A 5.3 Å resolution, cryo-electron microscopy (cryoEM) map of Chikungunya virus-like particles (VLPs) has been interpreted using the previously published crystal structure of the Chikungunya E1-E2 glycoprotein heterodimer. The heterodimer structure was divided into domains to obtain a good fit to the cryoEM density. Differences in the T = 4 quasi-equivalent heterodimer components show their adaptation to different environments. The spikes on the icosahedral 3-fold axes and those in general positions are significantly different, possibly representing different phases during initial generation of fusogenic E1 trimers. CryoEM maps of neutralizing Fab fragments complexed with VLPs have been interpreted using the crystal structures of the Fab fragments and the VLP structure. Based on these analyses the CHK-152 antibody was shown to stabilize the viral surface, hindering the exposure of the fusion-loop, likely neutralizing infection by blocking fusion. The CHK-9, m10 and m242 antibodies surround the receptor-attachment site, probably inhibiting infection by blocking cell attachment.

## Introduction

Chikungunya virus (CHIKV) is a mosquito-transmitted viral pathogen that causes fever, myalgia, rash, and severe arthritis in humans (*Powers and Logue, 2007*; *Simon et al., 2008*). The first reported human CHIKV infections occurred in East Africa in 1952 (*Robinson, 1955*). Prior to the 2005 epidemic on Réunion Island, CHIKV was not regarded as a highly prevalent virus. An adaptive mutation in the E1 protein (E1-A226V) that allowed CHIKV to replicate more efficiently in *Aedes albopictus* is considered to be the primary reason for its recent extensive spread, infecting millions of individuals in Africa and Asia (*Tsetsarkin et al., 2007*; *Kumar et al., 2008*; *Santhosh et al., 2008*). In some CHIKV-infected patients, severe damage in joint tissues can cause debilitating chronic arthritis. In the recent outbreaks, a change in pathogenesis was observed, with some cases progressing to fatal encephalitis. An autochthonous CHIKV outbreak in Italy in 2007 and the presence of the *Aedes albopictus* vector in many areas of Europe and the Americas have raised concern of further spread of the virus. Currently, there is no vaccine or antiviral agent approved for use in humans.

CHIKV belongs to the alphavirus genus of the *Togaviridae* family (*Kuhn, 2007*). Alphaviruses are a group of positive-sense, single-stranded RNA, enveloped viruses transmitted by arthropods (*Griffin, 2007*). The alphavirus genome encodes four non-structural and five structural proteins. The non-structural

**eLife digest** The Chikungunya virus is carried by mosquitos and can cause a number of diseases in humans including encephalitis, which can be fatal in some cases, and severe arthritis. A recent mutation in the E1 protein of the virus has allowed it to efficiently reproduce in a different species of mosquitos, leading to a Chikungunya epidemic in Réunion Island in 2005 and the subsequent infection of millions of individuals in Africa and Asia. The virus also has the potential to spread to many areas of Europe and the Americas.

Chikungunya virus has a single-stranded RNA genome that codes for four non-structural proteins and five structural proteins. Based on this knowledge it has been possible to develop virus-like particles that can be used to immunise non-human primates against Chikungunya infection by inducing antibody production. However, the development of vaccines for Chikungunya in humans will require a deeper understanding of how these antibodies produced by the vaccine interact with the virus and more detailed information about the structures of the virus and antibodies.

Sun *et al.* have used two techniques – X-ray crystallography and electron cryo-microscopy – to determine the structure of Chikungunya virus-like particles, and to obtain new insights into the interactions of these particles with four related antibodies. Electron cryo-microscopy was used to figure out the structure of the particles at near atomic resolution, and X-ray crystallography was used to determine the atomic resolution structures of two of the four Fab antibodies that neutralize the Chikungunya virus. Electron cryo-microscopy was also used to probe the complex formed by the interactions between the virus-like particles and the antibodies.

Sun *et al.* were able to identify the likely viral receptor site that is blocked by three of the antibodies when they neutralize the virus; the fourth antibody is thought to act by immobilizing one of the domains of protein E2, thereby hiding the "fusion loop" that allows the virus to enter and infect human tissue. It is hoped that these findings will contribute to efforts to combat the spread of the Chikungunya virus worldwide.

proteins are required for virus replication, protein modification, and immune antagonism. The structural proteins (capsid-E3-E2-6K-E1) are synthesized as a polyprotein from a subgenomic promoter, and are cleaved post-translationally into separate proteins by an autoproteinase and signalase. The E1 glycoprotein participates in cell fusion (*Lescar et al., 2001*), whereas the E2 glycoprotein binds to cellular receptors (*Smith et al., 1995*; *Zhang et al., 2005*) and initiates clathrin-dependent endocytosis (*Solignat et al., 2009*). Virus core assembly is initiated by interactions between the genomic RNA and the nuclear capsid protein (NCP) (*Tellinghuisen et al., 1999*; *Tellinghuisen and Kuhn, 2000*; *Linger et al., 2004*) in the cytoplasm. The E3 protein is essential for the proper folding of p62, the precursor to E2, and the formation of the p62-E1 heterodimer (*Mulvey and Brown, 1995*; *Carleton et al., 1997*). Although E3 remains part of mature Semliki Forest virus (SFV) and Venezuelan equine encephalitis virus (VEEV), apparently, it is not a component of mature CHIKV (*Simizu et al., 1984*). The small 6 kDa protein, 6K, associates with the p62-E1 heterodimer and is transported to the plasma membrane prior to assembly. The 6K protein facilitates particle morphogenesis but is not stoichiometrically incorporated into virions (*Gaedigk-Nitschko and Schlesinger, 1990*, *1991*).

Alphaviruses are icosahedral particles that have T = 4 quasi-icosahedral symmetry (*Paredes et al., 1993*; *Venien-Bryan and Fuller, 1994*; *Cheng et al., 1995*; *Zhang et al., 2002*, *2011*; *Kostyuchenko et al., 2011*). The ectodomain forms 80 spikes, each consisting of three copies of E1-E2 heterodimers. There are 20 icosahedral "i3" spikes, situated on icosahedral 3-fold axes, and 60 quasi-3-fold "q3" spikes in general positions containing a quasi-3-fold axis consistent with T = 4 symmetry (*Figure 1A*). Thus, each icosahedral asymmetric unit contains one complete q3 spike and one-third of an i3 spike. E1 of CHIKV is a 439 amino acid protein with an N-linked glycosylation site at residue Asn141. The ectodomain is formed by the 404 N-terminal residues, followed by a 30 residue transmembrane (TM) helix and a five amino acid cytoplasmic domain. The E1 ectodomain consists of three β-barrel domains (*Lescar et al., 2001*; *Gibbons et al., 2004*; *Li et al., 2010*; *Voss et al., 2010*). Domain I is spatially located between domains II and III with the fusion peptide lying at the distal end of domain II. E2 of CHIKV is 423 amino acids in length and has two N-linked glycans at positions Asn263 and Asn345. The E2 364 residue ectodomain is followed by

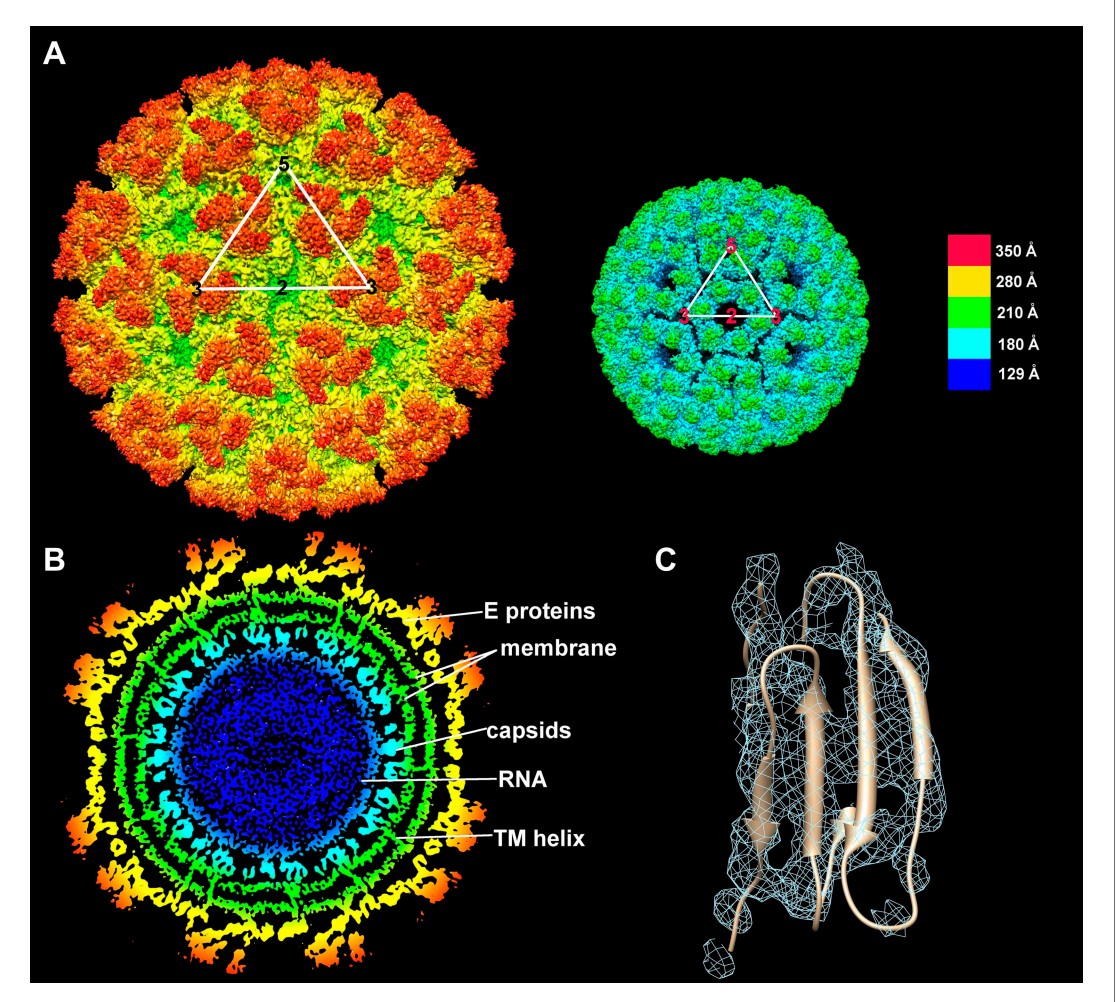

**Figure 1**. Structure of the CHIK VLPs. (**A**) Surface-shaded figure of ectodomain (left) and surface-shaded figure of nucleocapsid (right), colored according to the radial distance from the center of the virus. White triangles indicate one icosahedral asymmetric unit. (**B**) Cross-section of the virus showing density above 1.5 σ also colored according to the radial distance from the center of the virus. (**C**) Resolution of β-strands in the E1 domain III.

a 26-residue TM helix and a 33-residue cytoplasmic domain. The E2 ectodomain consists of three distinct immunoglobulin-fold domains with domain A, the putative receptor-binding domain, lying between domains B and C (*Li et al., 2010*; *Voss et al., 2010*). In the mature virus, domain B covers the fusion loop in domain II of E1. The virus becomes fusogenic in acid pH (*Wahlberg and Garoff, 1992*) when sets of three E1 molecules combine to make a fusogenic trimer after expelling the three E2 proteins from the center of the spike. The three fusion loops are then exposed outwards for insertion into the host cell membrane (*Gibbons et al., 2004*). Similar large conformational changes occur in other enveloped viruses (*Kielian and Rey, 2006*) although little is known of the structural intermediates through which these changes take place.

The nucleocapsid core consists of 30 hexamers around each 2-fold axis and 20 pentamers at each 5-fold vertex of the nuclear capsid (*Figure 1A*), and is separated from the ectodomains of E1 and E2 by a lipid membrane (*Figure 1B*). The N-terminal 118 amino acids of Sindbis (SINV) (*Choi et al., 1991*) and SFV (*Choi et al., 1997*) nuclear capsid contain positively charged residues that associate with the genomic RNA (*Zhang et al., 2002*). A similar positively charged N-terminal region exists in CHIKV NCP and functions to recognize the viral genome for packaging during NCP assembly (*Perera et al., 2001*; *Hong et al., 2006*). The carboxy-terminal region of the NCP has a chymotrypsin-like fold and acts as an auto-catalytic proteinase during processing of the structural polyprotein (*Hahn and Strauss, 1990*; *Choi et al., 1991*, *1997*). In addition, there is a hydrophobic pocket in the C-terminal

domain of the NCP that binds the 33 amino acid cytoplasmic tail of E2, effectively linking the proteins of the inner core with those on the surface of the virion (*Lee et al., 1996*; *Skoging et al., 1996*). This association of E2 with the nuclear capsid is essential for virus assembly (*Owen and Kuhn, 1997*).

Many studies have analyzed the interaction of antibodies with alphaviruses (*Meyer and Johnston, 1993*; *Griffin et al., 2001*; *Hunt et al., 2010*). Most neutralizing antibodies bind to E2, as this glycoprotein is more exposed on the viral surface than E1. The mode of neutralization differs considerably among antibodies, with some blocking attachment to cells and others inhibiting the conformational changes required for fusion (*Hernandez et al., 2008*). The only published structural studies of alphavirus-antibody complexes were a cryoEM analysis of Ross River virus complexed with an antibody that blocked receptor attachment (*Smith et al., 1995*) and a cryoEM investigation of SINV complexed with an antibody that inhibited the conformational changes required to form fusogenic E1 trimers (*Hernandez et al., 2008*).

Expression of the CHIKV structural C-E3-E2-6K-E1 polyprotein gives rise to virus-like particles (VLPs) that have been used to immunize and protect non-human primates against CHIKV infection (*Akahata et al., 2010*). Non-human primates immunized with these VLPs developed high-titer neutralizing antibodies that protected against viremia after high-dose challenges. Moreover, immunodeficient mice that were passively administered purified IgG from immunized non-human primates were protected against CHIKV challenge (*Couderc et al., 2009*). Previously, we published the structure of the CHIK VLPs at a resolution of 18 Å using cryoEM reconstruction of single particles (*Akahata et al., 2010*). Here, we report a 5.3 Å three-dimensional high-resolution reconstruction of the CHIK VLP. We also describe pseudo-atomic resolution structures of CHIK VLPs complexed with Fab fragments from four different neutralizing mouse monoclonal antibodies (MAb CHK-9, CHK-152, m10 and m242). These studies are the first structural investigations of CHIKV complexed with neutralizing MAbs, based on 15 Å resolution cryoEM reconstructions of the complexes, the 5.3 Å resolution structure of the VLPs and 3 Å resolution crystallographic studies of the Fab fragments. These structures suggest that MAb CHK-152 neutralizes infectivity by inhibiting fusion whereas the other MAbs probably neutralize by blocking attachment.

## Results and discussion

### CryoEM structure of CHIK VLPs

The structure of CHIK VLPs was determined to a resolution of ~5.3 Å. The trimeric appearance of individual spikes was recognizable immediately, along with the inclined positions of the three E1 molecules surrounding each spike. Further inspection of the E1 and E2 molecules showed β-barrel structures surrounding empty cavities, thus yielding a good indication of the positions of each domain. Although resolution of individual β-strands was possible only in some instances (*Figure 1C*), identification of individual side chains was not possible. The E3 glycoprotein, which is observed in some alphavirus structures (*Zhang et al., 2011*), was not apparent in the CHIK VLP cryoEM density map. Densities for the E2 B-domain were weak in all four E1-E2 heterodimers, consistent with the crystal structures of the CHIKV E1-E2 heterodimer (*Voss et al., 2010*) and the SINV trimeric spike (*Li et al., 2010*).

Within the CHIK VLP structure, the four quasi-equivalent chymotrypsin-like nucleocapsid proteins were organized as pentamers around the 5-fold vertices and hexamers around the 2-fold icosahedral symmetry axes (*Figure 1A*). The cryoEM density height of the nucleocapsid proteins was roughly 5% less than that of the glycoproteins. The glycoproteins and the nuclear capsid were separated by a ~45 Å-wide lipid membrane. The external and internal lipid leaflets had a density of about 5 σ, whereas the protein density of the glycoproteins and the NCPs had a density of about 7 σ. The two lipid leaflets were separated by a ~15 Å wide region, where the density was less than 1.5 σ, representing the loosely packed aliphatic chains of the lipid molecules (*Figure 1B*). The membrane was traversed by 60 × 4 pairs of α helices representing the E1 and E2 carboxy-terminal regions (*Figure 1B*). The quality of the membrane density in the icosahedrally averaged map demonstrates that the symmetry of the glycoproteins and nuclear capsid impose icosahedral symmetry on the flexible membrane by confining it to a limited space.

### Refinement and verification of the structure

A number of procedures have been developed for the refinement of atomic parameters when the resolution of the structural data is insufficient to follow the polypeptide chains with certainty or to

recognize the identity of amino acids (*Chapman, 1995*; *Volkmann and Hanein, 1999*; *Tama et al., 2004*; *Fabiola and Chapman, 2005*; *Goddard et al., 2007*; *Topf et al., 2008*; *Trabuco et al., 2008*; *Lasker et al., 2010*; *Schröder et al., 2010*; *Zhu et al., 2010*). These procedures increase the number of refinable parameters beyond those necessary to position and orient a rigid known structure into a relatively low-resolution cryoEM density map, thus reducing the ratio of observed data points (e.g. structure amplitudes) to refinable parameters. However severe restraints are placed on the geometry of the structure (thus providing additional observational data) while improving agreement between observed and calculated data in real or reciprocal space. Verification in reciprocal space can be achieved by tests on data that had been excluded in the refinement process, for instance by calculating R$_{free}$ (*Brünger, 1992*). Verification in real space can be established by observing anticipated structural features that had not been considered in the restrained refinement process.

In the analysis of the CHIKV VLPs described here, the initial interpretation was performed by fitting the structure of the CHIKV E1-E2 heterodimer (*Voss et al., 2010*) into the cryoEM map of the CHIKV VLPs assuming strict T = 4 symmetry between quasi-equivalent structures. Subsequently, the number of refinable parameters was increased by breaking the structure into domains, based on the assumption that the domains likely are more rigid than the hinges in the polypeptide chain that hold the domains together. In addition, constraints imposed by the assumption of strict T = 4 symmetry were abandoned, thus further multiplying the number of refinable parameters by four. Verification came from the improved correlation of the domain-fitted structure with the amino acid sequences, the improved correlation between density heights of quasi-equivalent parts of the structure with each other, and from the emergence of features such as the position of the N-linked carbohydrate moieties.

## Fitting of the CHIKV E1-E2 crystal structure into the cryoEM density map

In contrast to the interpretation of the cryoEM density of VEEV (*Zhang et al., 2011*), which had a resolution of 4.8 Å, we were able to use the crystallographically determined structure of the glycoproteins to aid in the analysis. The crystal structure of CHIKV E1-E2 (*Voss et al., 2010*) was fitted into the 5.3 Å cryoEM map using the program EMfit to maximize the average value of the density (sumf) at all atomic positions (*Table 1A*), to minimize the steric clash between symmetry related structures, and to minimize the percentage of atoms in low density (*Rossmann et al., 2001*). Initially, the E1-E2 heterodimer crystal structure was fitted as a rigid body generating the four quasi-equivalent structures at positions #1, #2, #3, and #4 (*Figure 2A*) while refining the best positions of the quasi-symmetry axes (see Materials and Methods), and using the icosahedral symmetry to generate the structure in neighboring icosahedral asymmetric units. The resultant locations of the four quasi-equivalent CHIKV heterodimers were used as starting positions for refinement of individual domains. The average height of the density at all fitted atoms (sumf) improved by ~12% for the independently fitted domains. The value of sumf was at least 38% lower for the B domains than for any of the other domains. This is consistent with the disorder of the B domain in the crystal structure of the SINV spike (*Li et al., 2010*) and the large temperature factors for the atoms in the B domain in the crystal structure of the CHIKV heterodimer (*Voss et al., 2010*). The height of the average value of the cryoEM density taken over all atomic positions within a molecular domain was higher at the quasi-equivalent position #4 than at the other three quasi-equivalent positions (*Table 1A*). This might be due to crowding around the 5-fold symmetry axes, which reduces the variability in the heterodimer position.

Once the individual domains were fitted (*Figure 3A*), the bond geometry between the carboxy end of one domain and the amino end of the next domain were regularized. The distance between these ends prior to regularization (*Table 2*) did not exceed 6.9 Å (excluding the flexible domain B), indicating that the quality of fit was reasonable. Comparison of the hinge angles between domains in the four quasi-equivalent positions (*Table 3*) did not exceed 13°, except for defining the orientation of the B domain in E2. Considering that the length of the longest domain (DII of E1) is about 50 Å, this implies a relative difference of about 6 Å for atoms furthest from the hinge, corresponding to a maximum movement of about three standard deviations between superimposed structures.

After independent fitting of the domains, the root mean square (rms) distance between equivalent Cα atoms when making pairwise comparisons between E1-E2 heterodimers ranged from 0.5 Å to 2.4 Å (*Table 4*). This showed that the structures of the four quasi-equivalent parts of the model were more alike to each other than to the crystal structure of the CHIKV heterodimer (*Voss et al., 2010*). As such similarity is unlikely to occur by accident, this observation helped to confirm the validity of the process used to interpret the cryoEM map. The systematic change of the E1-E2 heterodimers in the VLPs

**Table 1.** Quality of fitting the atomic structural fragments into the cryoEM density

**A) Average height of the densities at the atomic positions (sumf) (*Rossmann et al., 2001*) on fitting the E1E2 heterodimer CHIKV structure into the cryoEM map at four quasi equivalent positions**

| All atoms | | T = 4 fitting | | | | | Independent domain fitting | | | | |
|---|---|---|---|---|---|---|---|---|---|---|---|
| | | #1 | #2 | #3 | #4 | Average | #1 | #2 | #3 | #4 | Average |
| | I | 16.1 | 15.8 | 17.0 | 17.7 | 16.6 | 18.2 | 17.0 | 18.5 | 19.9 | 18.5 |
| E1* | II | 15.7 | 17.2 | 18.8 | 16.7 | 17.0 | 18.0 | 18.6 | 19.0 | 18.5 | 18.6 |
| | III | 15.7 | 14.2 | 15.2 | 18.0 | 15.7 | 18.7 | 15.9 | 18.2 | 20.7 | 18.3 |
| | A | 15.5 | 18.0 | 19.0 | 17.5 | 17.4 | 18.8 | 17.8 | 19.9 | 19.4 | 18.8 |
| E2* | B | 10.0 | 9.4 | 9.6 | 9.6 | 9.8 | 10.9 | 11.3 | 11.7 | 12.4 | 11.7 |
| | C | 15.3 | 15.4 | 19.0 | 19.1 | 16.8 | 21.0 | 19.9 | 20.2 | 22.4 | 20.5 |
| | D | 13.3 | 16.4 | 15.8 | 16.1 | 15.4 | 16.4 | 17.2 | 16.0 | 16.8 | 16.5 |
| Average | | 14.9 | 15.8 | 17.0 | 16.8 | 16.1 | 17.8 | 17.3 | 18.2 | 19.0 | 18.1 |
| **Main chain atoms only** | | **T = 4 fitting** | | | | | **Independent domain fitting** | | | | |
| | | #1 | #2 | #3 | #4 | Average | #1 | #2 | #3 | #4 | Average |
| | I | 20.5 | 18.4 | 21.5 | 22.9 | 20.8 | 23.2 | 21.6 | 22.6 | 26.3 | 23.4 |
| E1* | II | 17.0 | 20.9 | 23.0 | 20.2 | 20.3 | 21.7 | 23.3 | 23.5 | 22.8 | 22.8 |
| | III | 18.9 | 15.3 | 17.5 | 22.3 | 18.5 | 22.3 | 17.9 | 22.6 | 26.6 | 22.4 |
| | A | 16.5 | 21.1 | 22.2 | 19.4 | 19.8 | 22.0 | 20.3 | 23.4 | 22.6 | 22.1 |
| E2* | B | 10.7 | 9.4 | 9.9 | 10.1 | 10.0 | 10.8 | 11.2 | 13.0 | 13.4 | 12.1 |
| | C | 18.7 | 19.5 | 27.4 | 26.9 | 23.1 | 27.7 | 27.1 | 27.4 | 30.8 | 28.3 |
| | D | 13.4 | 18.7 | 18.5 | 18.5 | 17.3 | 19.4 | 19.8 | 19.1 | 19.0 | 19.3 |
| Average | | 16.5 | 17.6 | 20.0 | 20.0 | 18.5 | 21.0 | 20.2 | 21.7 | 23.1 | 21.5 |

**B) Average height of the densities at the atomic positions (sumf) (*Rossmann et al., 2001*) on fitting the CHIKV homology model of the capsid protein into the cryoEM map at the four quasi equivalent positions**

| All atoms | T = 4 fitting | | | | | Independent domain fitting | | | | |
|---|---|---|---|---|---|---|---|---|---|---|
| | #1 | #2 | #3 | #4 | Average | #1 | #2 | #3 | #4 | Average |
| D1† | 14.9 | 13.3 | 13.9 | 16.1 | 14.6 | 16.1 | 14.6 | 15.4 | 17.3 | 15.8 |
| D2† | 17.0 | 17.4 | 18.1 | 18.1 | 17.6 | 18.3 | 19.1 | 17.4 | 19.0 | 18.6 |
| Average | 16.0 | 15.4 | 16.0 | 17.2 | 16.1 | 17.4 | 17.1 | 16.5 | 18.2 | 17.3 |
| **Main chain atoms only** | **T = 4 fitting** | | | | | **Independent domain fitting** | | | | |
| | #1 | #2 | #3 | #4 | Average | #1 | #2 | #3 | #4 | Average |
| D1† | 18.5 | 16.5 | 17.8 | 18.9 | 17.9 | 20.4 | 16.9 | 19.2 | 22.7 | 19.8 |
| D2† | 21.3 | 21.5 | 21.4 | 23.2 | 21.9 | 22.8 | 24.2 | 20.4 | 25.3 | 23.2 |
| Average | 19.9 | 19.0 | 19.6 | 21.1 | 19.9 | 21.8 | 21.0 | 19.9 | 24.1 | 21.6 |

**C) Average height of the densities at the atomic positions (sumf) (*Rossmann et al., 2001*) on fitting the CHIKV model of the transmembrane protein into the cryoEM map at the four quasi equivalent positions**

| All atoms | T = 4 fitting | | | | | Independent domain fitting | | | | |
|---|---|---|---|---|---|---|---|---|---|---|
| | #1 | #2 | #3 | #4 | Average | #1 | #2 | #3 | #4 | Average |
| E1‡ | 13.7 | 14.6 | 13.2 | 18.6 | 15.0 | 13.9 | 14.4 | 14.7 | 17.5 | 15.1 |
| E2‡ | 14.7 | 12.8 | 11.8 | 17.5 | 14.2 | 15.1 | 14.1 | 13.7 | 18.9 | 15.5 |
| Average | 14.4 | 13.5 | 12.3 | 17.9 | 14.6 | 14.5 | 14.2 | 14.2 | 18.2 | 15.3 |
| **Main chain atoms only** | **T = 4 fitting** | | | | | **Independent domain fitting** | | | | |
| | #1 | #2 | #3 | #4 | Average | #1 | #2 | #3 | #4 | Average |
| E1‡ | 16.2 | 17.6 | 16.0 | 24.1 | 18.5 | 15.7 | 17.3 | 16.7 | 23.9 | 18.4 |
| E2‡ | 16.7 | 15.0 | 13.6 | 20.7 | 16.5 | 17.7 | 16.7 | 15.8 | 23.7 | 18.5 |
| Average | 16.5 | 16.0 | 14.4 | 22.0 | 17.5 | 16.7 | 17.0 | 16.2 | 23.8 | 18.4 |

*Table 1. Continued on next page*

*Table 1. Continued*

*Domain definition: E1 I (residues 1-36, 132-168, 273-293), II (residues 37-131, 169-272), III (residues 294-393). E2 A (residues 16-134), B (residues 173-231), C (residues 269-342), D (residues 7-15, 135-172, 232-268).
†Domain definition: D1(residues 119 to 183), D2(residues 184 to 267).
‡Domain definition: E1 (residues 394-439), E2(residues 343-423).

compared to the crystal structure is the result of accommodating the different quasi-equivalent environments. In contrast to the results for the related alphavirus, VEEV (*Zhang et al., 2011*), the presence of differences larger than the standard deviation between equivalenced Cα atoms between the four quasi-equivalent structural components of the CHIKV VLPs explains why averaging of the quasi equivalent density did not improve the map.

The quality of the side chain placements was determined by correlating their densities in the four quasi-equivalent positions. The density associated with each residue was taken as the average density of all the non-hydrogen atoms in that residue. A high correlation coefficient (see Materials and Methods for definition) indicates that the variation of cryoEM density along each polypeptide chain was similar. As the average density for each residue included all atoms in the side chain, a high correlation also implied similarity of the side chain densities. These correlation coefficients (*Table 5A*) varied from 0.44 to 0.30, which was significantly greater than the values (0.40–0.08) for the initial T = 4 rigid body fitting. Furthermore, there was a reasonable correlation of the average crystallographic temperature factors for each amino acid residue of the CHIKV heterodimer crystal structure (*Voss et al., 2010*) with the height of the cryoEM density at each amino acid position (*Table 5A*).

## The N-linked carbohydrate sites

The fitting of the independent domains of the CHIKV heterodimer into the cryoEM map was further validated by finding the densities associated with N-linked glycans, after setting to zero all densities at grid points within 3.0 Å of every fitted atom. The remaining densities should correspond to the sugar moieties on E1 and E2 (*Pletnev et al., 2001*). The shape and size of the carbohydrate moiety for each of the three N-linked glycosylation sites is consistent in each of the four quasi-equivalent positions. The average volume of the carbohydrates at the four E1 (Asn141) sites in DI of E1 is 258 Å$^3$ with an average height of 2.2 σ, with most of the noise density not being greater than 1.0 σ. This would correspond to about one sugar moiety. The densities corresponding to the four quasi-equivalent E2 (Asn263) sites in the E2 connecting ribbon were somewhat fragmented, making it difficult to determine their volumes. However, at lower (15 and 10 Å) resolution, the densities were well defined. The volume of the density and average height associated with the four quasi-equivalent sites at E2 (Asn345) in domain C of E2 were as large as the sites at E1 (Asn141), but because of their proximity to the viral membrane, their boundaries were uncertain. The centers of the carbohydrate moieties were ~6 Å from the N atom of the glycosylated Asn residues.

## Interactions between the E1 and E2 glycoproteins

Contacts between the E1 and E2 glycoproteins were determined (Materials and Methods) for the T = 4 fitted structure and compared with the improved, independently fitted, domain structure (*Table 6*). The E1 and E2 glycoproteins that form the four heterodimers in the VLPs have more contacts with each other in the independently fitted structure than in the crystallographic heterodimer structure. Particularly striking was the increase in hydrophobic contacts in the independently fitted results, which demonstrated how the heterodimer adapted itself to the quasi-equivalent symmetry of the VLPs compared to the constraints imposed by the crystal lattice.

Unlike the T = 4 fitted structure, the independently fitted structure lacked exact symmetry between the i3 and q3 spikes. Nevertheless, the q3 spike had apparent, although not exact, 3-fold symmetry within itself and with its surroundings. A comparison of the i3 and q3 structures showed that the E1 molecules wrap around the central E2 molecules more tightly in the i3 spike as indicated by the increased number of interactions between the E2 molecules in the i3 spike compared to the q3 spike. Thus, the i3 trimers in the VLPs were more compact with a greater number of interactions between its component molecules than the q3 trimer. The greater compactness of the i3 spike was confirmed by determining the radius of gyration of the i3 and the q3 spikes as a function of the radial distance from

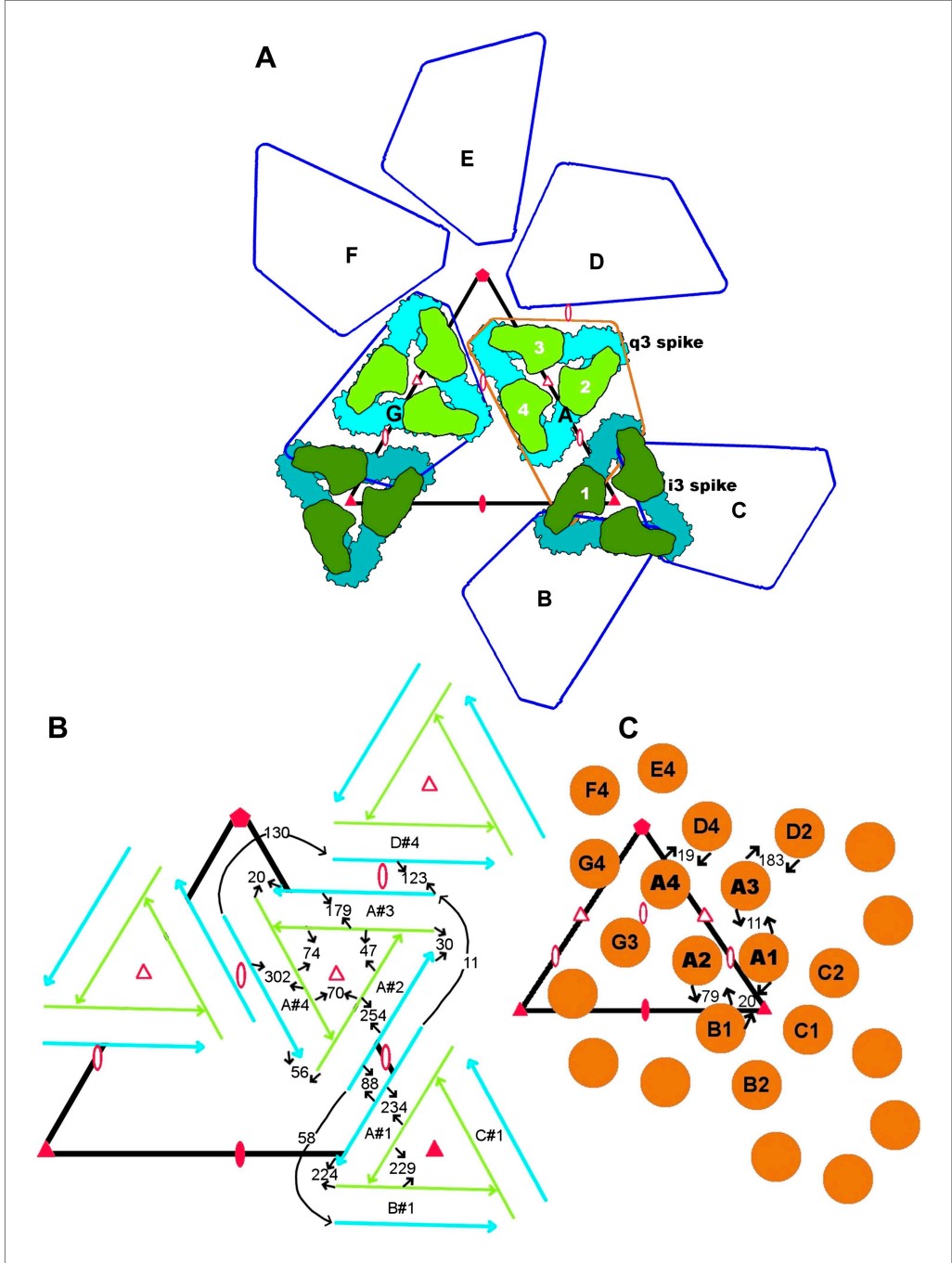

**Figure 2**. Diagrammatic representation of contacts between amino acid residues in adjacent subunits.
(**A**) Diagrammatic organization of the E1 and E2 subunits according to T = 4 icosahedral symmetry. The white
numbers show the sequence in which the four independent subunits were generated. The black capital letters
indicate the sequence of generating seven of the icosahedral asymmetric units. Icosahedral symmetry
elements are shown as filled triangles and ellipses. Quasi-symmetry elements are shown as red outlined
triangles and ellipses. (**B**) Contacts between the E1 (blue) and E2 (green) molecules. (**C**) Contacts between NCPs.
(**B**) and (**C**) show the number of contacts between the indicated molecules. The quasi T = 4 related positions #1, #2,
#3 and #4 are indicated, prefixed by their icosahedral symmetry identification A, B, C and D. The center of the "i3"
icosahedral spike is indicated by a filled red triangle and the center of the "q3" quasi-3-fold spike is indicated by
an outlined red triangle.

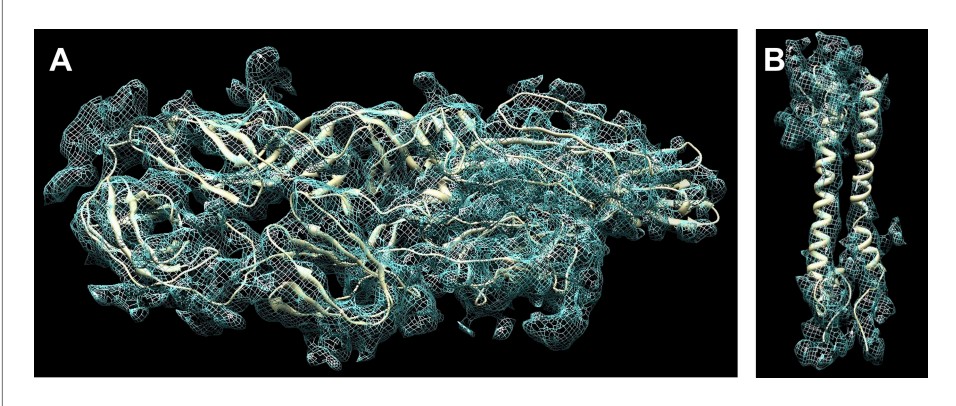

**Figure 3**. Fit of the atomic structure into the cryoEM density. (**A**) Fit of the E1E2 heterodimer, (**B**) Fit of the capsid protein. The quasi equivalent subunit closest to the icosahedral 5-fold axis was chosen for display.

the viral center (Materials and Methods). At the base of the spike, where the E1 molecules make contacts between spikes, the radius of gyration was up to 5.8 Å less for the i3 spike than for the q3 spike (*Table 7*). At the top of the spike, the i3 and q3 spikes showed less difference. In addition, the whole of the q3 spike was translated away from the center of the VLPs by ~1.8 Å. Thus, in comparison with the 20 i3 spikes, the 60 q3 spikes may be poised to release the E2 molecules from the center of the spikes in preparation for the formation of the E1 fusogenic homotrimers early in the infection (*Li et al., 2010*).

Prior studies (*Li et al., 2010*) have suggested that the trigger for movement of domain B to uncover the fusion loop on E1 at low pH might be the protonation of the highly conserved His169 and His256 residues in SINV on the β-ribbon that connects the ends of the B domain to the A and C domains of E2 (*Li et al., 2010*). In the CHIK VLP structure, His170 (equivalent to SINV His 169) does not interact closely with other negatively charged residues that would be required for a pH-sensing switch whereas CHIKV His 256 (equivalent to SINV His256) contacts Glu166.

## Fitting of a homology model of CHIKV NCP structure into the cryoEM density map

The crystal structures of SINV (*Choi et al., 1991*; *Tong et al., 1992*) and SFV (*Choi et al., 1997*) NCPs had been determined previously. Of these, the amino acid sequence of the NCP of SFV has greater

**Table 2.** Distances in Å between N and C termini of fitted domains before regularization

| Domains | | Quasi equivalent positions | | | |
|---|---|---|---|---|---|
| | | #1 | #2 | #3 | #4 |
| | I 36 to II 37 | 4.6 | 5.6 | 5.5 | 3.5 |
| | II 131 to I 132 | 4.8 | 4.1 | 4.4 | 4.1 |
| E1 | I 168 to II 169 | 2.8 | 6.4 | 5.4 | 5.1 |
| | II 272 to I 273 | 5.8 | 4.7 | 3.9 | 2.1 |
| | I 293 to III 294 | 5.4 | 6.9 | 3.7 | 5.0 |
| | D 15 to A 16 | 3.8 | 6.3 | 5.7 | 5.9 |
| E2 | A134 to D135 | 3.1 | 5.0 | 4.4 | 2.3 |
| | D172 to B173 | 3.4 | 8.8 | 3.0 | 5.7 |
| | B231 to D232 | 6.2 | 7.7 | 5.9 | 7.2 |
| NCP | D268 to C269 | 5.1 | 5.2 | 4.1 | 4.0 |
| | (D1)183 to (D2)184 | 1.2 | 2.2 | 1.6 | 1.6 |

**Table 3.** Hinge angle change between domains in different heterodimers

**Hinge angle change in E1 between domain I and domain II (in degrees)**

|  | Position #1 | Position #2 | Position #3 | Position #4 |
| --- | --- | --- | --- | --- |
| Crystal structure | -14.4 | -6.6 | 7.5 | -8.3 |
| Position #1 |  | 8.0 | 9.3 | 6.8 |
| Position #2 |  |  | 3.1 | 2.1 |
| Position #3 |  |  |  | -3.3 |

**Hinge angle change in E1 between domain I and domain III (in degrees)**

|  | Position #1 | Position #2 | Position #3 | Position #4 |
| --- | --- | --- | --- | --- |
| Crystal structure | -7.3 | -8.0 | -9.4 | 2.1 |
| Position #1 |  | -3.5 | 10.3 | 7.7 |
| Position #2 |  |  | 13.0 | 9.2 |
| Position #3 |  |  |  | 7.6 |

**Hinge angle change in E2 between domain A and domain B (in degrees)**

|  | Position #1 | Position #2 | Position #3 | Position #4 |
| --- | --- | --- | --- | --- |
| Crystal structure | -8.4 | 10.4 | 22.2 | -22.3 |
| Position #1 |  | -14.1 | 25.5 | -29.7 |
| Position #2 |  |  | 26.6 | -28.6 |
| Position #3 |  |  |  | -26.3 |

**Hinge angle change in E2 between domain A and domain C (in degrees)**

|  | Position #1 | Position #2 | Position #3 | Position #4 |
| --- | --- | --- | --- | --- |
| Crystal structure | -11.5 | -10.4 | -7.6 | -17.1 |
| Position #1 |  | -4.8 | -8.6 | -7.3 |
| Position #2 |  |  | 7.2 | 8.3 |
| Position #3 |  |  |  | -14.8 |

similarity to CHIKV. Therefore, a homology model for CHIKV NCP (residues 119 to 267) was generated based on the crystal structure of the SFV NCP (Protein Data Bank accession number 1VCP). This homology structure was fitted into the CHIK VLP cryoEM density map by assuming T = 4 symmetry, using the EMfit program (*Table 1B*). The positions of the quasi-symmetry axes were refined to optimize the fit to the density and to minimize clashes between symmetry related NCP structures. The refinement found that the placement of the quasi-symmetry axes did not change significantly from those optimized while fitting the glycoproteins. The orientation and placement of the homology NCP in the cryoEM density was essentially the same as described for other alphaviruses (*Zhang et al., 2002*). Furthermore, the slight asymmetric twist of the pentameric and hexameric rings about the 5 and 2-fold axes, respectively (*Figures 1A,2B*) was the same as that observed for Ross River virus (*Cheng et al., 1995*).

The lack of a substantial rigid association between the two lobes of the NCP's chymotrypsin-like structure suggests that there could be conformational adjustments between the two lobes. Consequently, the NCP structure was split into two domains (amino acid residues 119 to 183 and 184 to 267) representing the two lobes on either side of the substrate-binding cleft. These domains were then fitted independently at each of the four quasi-equivalent positions. The sumf values (*Table 1B*) for fitting the NCP into the pentameric capsomer density (position #4, *Figure 3B*) were slightly greater than for the NCPs in the hexameric capsomer (positions #1, #2, and #3), as was the case for the E1 protein fitting near the 5-fold axes. The correlation of the average residue densities between the four quasi-equivalent NCP positions was roughly the same as for fitting of the glycoproteins (*Table 5*), indicating the same level of amino acid recognition for the glycoproteins and the NCPs.

Contrary to the organization of the E1-E2 glycoproteins, where there are extensive inter-heterodimer contacts within the trimeric spikes, there were few contacts among the three NCP molecules around the icosahedral 3-fold axes and no contacts among the three NCP molecules

**Table 4.** RMS deviation in Å between quasi equivalent Cα atoms

| E1 | #1 | #2 | #3 | #4 | Xtal |
|---|---|---|---|---|---|
| #1 | 0.00 | 1.21 | 1.25 | 1.08 | 1.69 |
| #2 | | 0.00 | 1.28 | 1.27 | 1.52 |
| #3 | | | 0.00 | 1.10 | 1.68 |
| #4 | | | | 0.00 | 1.17 |
| Xtal | | | | | 0.00 |
| **E2** | **#1** | **#2** | **#3** | **#4** | **Xtal** |
| #1 | 0.00 | 2.18 | 1.79 | 1.90 | 1.50 |
| #2 | | 0.00 | 2.34 | 2.38 | 2.17 |
| #3 | | | 0.00 | 2.28 | 1.84 |
| #4 | | | | 0.00 | 1.92 |
| Xtal | | | | | 0.00 |
| **E1&E2** | **#1** | **#2** | **#3** | **#4** | **Xtal** |
| #1 | 0.00 | 1.94 | 1.65 | 1.68 | 2.28 |
| #2 | | 0.00 | 2.16 | 2.14 | 2.24 |
| #3 | | | 0.00 | 1.87 | 2.37 |
| #4 | | | | 0.00 | 1.88 |
| Xtal | | | | | 0.00 |
| **Capsid** | **#1** | **#2** | **#3** | **#4** | **Xtal** |
| #1 | 0.00 | 0.49 | 0.73 | 0.71 | 0.48 |
| #2 | | 0.00 | 1.09 | 0.94 | 0.71 |
| #3 | | | 0.00 | 0.63 | 0.56 |
| #4 | | | | 0.00 | 0.52 |
| Xtal | | | | | 0.00 |

around the quasi-3-fold axes. However, the six NCP molecules that form the hexameric capsomers around the icosahedral 2-fold axes made extensive contacts (*Figure 3B* and *Table 6*). These interactions were between the loop formed by residues 172 to 183 in one NCP molecule with residues in loops 190 to 206, 239 to 243 and 261 to 262 in a neighboring NCP molecule. Moreover, salt bridges were formed between Lys178 and Glu240. The NCP molecules around the icosahedral 5-fold axes made similar contacts and salt bridges, as expected by their quasi-similar environments.

It was unexpected that the positions of the quasi-2- and 3-fold symmetry elements were matched almost perfectly between the external organization of the glycoproteins into 80 trimers and the internal organization of the NCP into 12 pentamers and 30 hexamers (*Figure 2*). The nucleoplasmic cores assemble in the cytoplasm (*Weiss et al., 1989*; *Owen and Kuhn, 1996*; *Soonsawad et al., 2010*) and are transported to the plasma membrane where they interact with the endodomain of the E2 protein. This provides an opportunity for the C-terminal regions of E2 to bind to the hydrophobic pocket on the surface of the nucleocapsid (*Lee et al., 1996*), thus nucleating the E1-E2 glycoproteins to form an icosahedral shell in synchrony with the icosahedral symmetry of the cores.

## Model building of the E1-E2 TM region and endodomains

The crystal structure of CHIKV E1-E2 (*Voss et al., 2010*) consists of residues 1-393 of E1 and 7-342 of E2 but does not include the TM regions and endodomains. Cα models were built of the missing part of the E1-E2 ectodomains, the TM helices, and the E2 endodomain, based on Rosetta structural predictions (*Lange and Baker, 2012*) and on the cryoEM density. The alignment of the amino acid sequence with the cryoEM density was based on assigning the kink observed in the TM helix of E1 to the likely flexible Thr-Gly-Gly sequence. In addition, the alignment was based on placing the interaction of Tyr 400 into the hydrophobic pocket around Trp 251 in the NCP, corresponding to SINV NCP residue Trp 247 (*Lee et al., 1996*). The TM regions of E1 and E2 were built independently for each of the four quasi-equivalent regions of the cryoEM map (*Figure 3C*). The E1 and E2 helices were parallel and situated close together with some hydrophobic contacts between them (*Table 6*). The T = 4 quasi-symmetry operators that relate the pairs of E1-E2 TM helices to other E1-E2 TM helical pairs were in similar positions to the symmetry operators that relate E1-E2 ectodomains to each other.

Although the lipid membrane is presumably flexible and likely to have different organization of lipid molecules in different particles, the cryoEM density showed a preference for roughly radial density features. Presumably, these densities represent an average of similarly placed lipid molecules, as was observed in the 4 Å resolution X-ray crystallographic map of the dsDNA PRD1 bacteriophage (*Cockburn et al., 2004*).

## Comparison with the VEEV structure

The only other high resolution structure of an alpha virus is that of VEEV (*Zhang et al., 2011*). One way of comparing the structure of VEEV with the structure of CHIK VLPs is by superimposing their icosahedral symmetry elements. This shows that the overall radius of these two viruses is essentially

**Table 5.** Correlation between amino acid sequence and cryoEM density

**A) Correlation between cryoEM densities in the four quasi equivalent positions of the E1 and E2 glycoproteins.**

| All atoms | T = 4 fitting | | | | | Independent domain fitting | | | | |
| --- | --- | --- | --- | --- | --- | --- | --- | --- | --- | --- |
| | 1 | 2 | 3 | 4 | B | 1 | 2 | 3 | 4 | B |
| 1 | 1.00 | 0.17 | 0.23 | 0.08 | 0.22 | 1.00 | 0.40 | 0.44 | 0.40 | 0.21 |
| 2 | | 1.00 | 0.28 | 0.38 | 0.30 | | 1.00 | 0.41 | 0.30 | 0.28 |
| 3 | | | 1.00 | 0.40 | 0.28 | | | 1.00 | 0.36 | 0.28 |
| 4 | | | | 1.00 | 0.19 | | | | 1.00 | 0.15 |
| B | | | | | 1.00 | | | | | 1.00 |

| Main chain atoms only | T = 4 fitting | | | | | Independent domain fitting | | | | |
| --- | --- | --- | --- | --- | --- | --- | --- | --- | --- | --- |
| | 1 | 2 | 3 | 4 | B | 1 | 2 | 3 | 4 | B |
| 1 | 1.00 | 0.11 | 0.20 | 0.11 | 0.19 | 1.00 | 0.40 | 0.41 | 0.41 | 0.22 |
| 2 | | 1.00 | 0.31 | 0.30 | 0.30 | | 1.00 | 0.36 | 0.32 | 0.26 |
| 3 | | | 1.00 | 0.41 | 0.27 | | | 1.00 | 0.37 | 0.26 |
| 4 | | | | 1.00 | 0.16 | | | | 1.00 | 0.12 |
| B | | | | | 1.00 | | | | | 1.00 |

**B) Correlation between cryoEM densities of the four quasi equivalent capsid proteins.**

| All atoms | T = 4 fitting | | | | Independent domain fitting | | | |
| --- | --- | --- | --- | --- | --- | --- | --- | --- |
| | #1 | #2 | #3 | #4 | #1 | #2 | #3 | #4 |
| #1 | 1.00 | 0.41 | 0.18 | 0.17 | 1.00 | 0.20 | 0.19 | 0.30 |
| #2 | | 1.00 | 0.26 | 0.25 | | 1.00 | 0.45 | 0.42 |
| #3 | | | 1.00 | 0.13 | | | 1.00 | 0.41 |
| #4 | | | | 1.00 | | | | 1.00 |

| Main chain atoms only | T = 4 fitting | | | | Independent domain fitting | | | |
| --- | --- | --- | --- | --- | --- | --- | --- | --- |
| | #1 | #2 | #3 | #4 | #1 | #2 | #3 | #4 |
| #1 | 1.00 | 0.37 | 0.16 | 0.26 | 1.00 | 0.22 | 0.17 | 0.33 |
| #2 | | 1.00 | 0.31 | 0.27 | | 1.00 | 0.34 | 0.43 |
| #3 | | | 1.00 | 0.15 | | | 1.00 | 0.31 |
| #4 | | | | 1.00 | | | | 1.00 |

**C) Correlation between cryoEM densities of the four quasi equivalent E1&E2 TM and endodomain regions.**

| All atoms | T = 4 fitting | | | | Independent domain fitting | | | |
| --- | --- | --- | --- | --- | --- | --- | --- | --- |
| | #1 | #2 | #3 | #4 | #1 | #2 | #3 | #4 |
| #1 | 1.00 | 0.43 | 0.17 | 0.41 | 1.00 | 0.37 | 0.34 | 0.35 |
| #2 | | 1.00 | 0.24 | 0.38 | | 1.00 | 0.44 | 0.34 |
| #3 | | | 1.00 | 0.20 | | | 1.00 | 0.46 |
| #4 | | | | 1.00 | | | | 1.00 |

| Main chain atoms only | T = 4 fitting | | | | Independent domain fitting | | | |
| --- | --- | --- | --- | --- | --- | --- | --- | --- |
| | #1 | #2 | #3 | #4 | #1 | #2 | #3 | #4 |
| #1 | 1.00 | 0.28 | 0.08 | 0.27 | 1.00 | 0.23 | 0.11 | 0.26 |
| #2 | | 1.00 | 0.17 | 0.43 | | 1.00 | 0.40 | 0.36 |
| #3 | | | 1.00 | 0.14 | | | 1.00 | 0.50 |
| #4 | | | | 1.00 | | | | 1.00 |

**Table 6.** Inter-molecular contacts (See methods for descriptions)

| Spike position | | Total | Hydrophobic | Possible H bonds | Possible salt bridges |
|---|---|---|---|---|---|
| **A) Number of atom-to-atom contacts between E1-E2 in heterodimer** | | | | | |
| i3 | #1 | 234 | 99 (42%) | 34 (13%) | 0 |
| q3 | #2 | 254 | 112 (44%) | 25 (10%) | 0 |
| | #3 | 179 | 59 (33%) | 33 (18%) | 0 |
| | #4 | 302 | 118 (39%) | 39 (13%) | 4 (1%) |
| Crystal | | 86 | 1 (1%) | 32 (37%) | 1 (1%) |
| **B) Number of atom-to-atom contacts of E2 with other E2s within a spike** | | | | | |
| i3 | #1 | 229 | 46 (20%) | 30 (13%) | 82 (36%) |
| q3 | #2 | 70 | 8 (11%) | 10 (14%) | 33 (47%) |
| | #3 | 47 | 14 (30%) | 6 (13%) | 10 (21%) |
| | #4 | 74 | 118 (23%) | 24 (46%) | 7 (9%) |
| **C) Number of atom-to-atom contacts between glycoprotein spikes** | | | | | |
| q3 to q3 | A#3 - D #4 | 123 | 42 (34%) | 21 (17%) | 7 (5%) |
| | A#4 - D#4 | 130 | 52 (40%) | 18 (14%) | 1 (1%) |
| i3 to q3 | A#2 - B#1 | 58 | 27 (47%) | 7 (12%) | 0 |
| | A#1 - A#2 | 88 | 29 (33%) | 19 (22%) | 0 |
| | A#1 - A#3 | 11 | 8 (73%) | 2 (8%) | 0 |
| **D) Number of atom-to-atom contacts between E1 and E2 within the spikes.** | | | | | |
| q3 | A#2 - A#3 | 30 | 16 (34%) | 5 (17%) | 0 |
| | A#3 - A#4 | 20 | 8 (40%) | 3 (14%) | 0 |
| i3 | A#4 - A#2 | 56 | 25 (47%) | 11 (12%) | 0 |
| | A#1 - B#1 | 224 | 112 (50%) | 21 (9%) | 22 (10%) |
| **E) Number of atom-to-atom contacts between capsid proteins*** | | | | | |
| | A#2 - B#1 | 79 | 20 (25%) | 13 (16%) | 15 (19%) |
| | A#1 - A#3 | 11 | 3 (27%) | 1 (9%) | 3 (27%) |
| | A#3 - D#2 | 183 | 51 (28%) | 22 (12%) | 38 (21%) |
| | A#4 - D#4 | 19 | 5 (26%) | 2 (11%) | 8 (42%) |
| | A#1 - B#1 | 20 | 9 (45%) | 2 (10%) | 2 (10%) |
| **F) Number of atom-to-atom contacts between E1 & E2 in TM region** | | | | | |
| | #1 | 100 | 60 (60%) | 3 (3%) | 0 |
| | #2 | 82 | 43 (52%) | 8 (10%) | 0 |
| | #3 | 20 | 12 (60%) | 1 (5%) | 0 |
| | #4 | 28 | 24 (86%) | 0 | 0 |

*See Figure 2.

the same. However, the positions and orientations of the molecular protein components have changed by up to 9.7 Å and 11.7°, respectively (*Table 8*), which is far greater than the RMS Cα atom differences between quasi equivalent positions of the component molecular components in the CHIK VLPs (maximum RMS distance is 2.4 Å, *Table 4*). It is also much larger than the RMS difference between Cα atoms on superimposing the equivalent molecular components of VEEV and CHIK VLPs without regard for the position and orientation of the icosahedral symmetry axes (*Table 8*). Thus the change in molecular positions and orientation with respect to the icosahedral framework is highly significant. Although the capsid protein amino acids are more conserved than any of the other structural proteins (*Table 8*), by far the largest differences in position and orientation of the molecular components occur for the capsid proteins (*Table 8*). The greater conservation of the amino acid sequences for the capsid protein

**Table 7.** Spike radii of gyration

| Distance from particle center (Å) | Radii of gyration (Å) | | | # of atoms | | |
|---|---|---|---|---|---|---|
| | i3 | q3 | SINV* crystal | i3 | q3 | SINV* crystal |
| 255 | 58.3 | 61.0 | 63.3 | 42 | 34 | 62 |
| 265 | 53.5 | 57.3 | 54.9 | 131 | 126 | 118 |
| 275 | 41.1 | 46.9 | 42.6 | 125 | 122 | 101 |
| 285 | 30.1 | 33.1 | 30.1 | 84 | 90 | 80 |
| 295 | 33.2 | 34.2 | 33.8 | 85 | 80 | 91 |
| 305 | 33.3 | 31.9 | 31.3 | 90 | 89 | 82 |
| 315 | 30.7 | 31.2 | 24.5 | 70 | 80 | 92 |
| 325 | 26.7 | 25.9 | | 41 | 47 | |
| 335 | 31.0 | 29.0 | | 1 | 1 | |
| Overall | 40.5 | 42.4 | 41.9 | 669 | 669 | 626 |

*Sindbis virus (SINV)

is also reflected in a lower RMS differences between Cα atoms between superimposed capsid structures of VEEV and CHIK VLPs (less than 1.2 Å) as opposed to a comparison of the other molecular components (greater than 2.0 Å, *Table 8*).

The above analysis shows that these two alpha-viruses maintain a fairly constant tertiary structure, notwithstanding moderate amino acid sequence variations. However, the quaternary organization has greater variability, although still retaining the same T = 4 quasi symmetry. Considering that the nucleocapsid core is assembled first and the subsequent association of the glycoproteins is based on the pre-existing cores, it is surprising that the difference of structure between the CHIKV and VEEV cores is less evident in the virus' ectodomains.

## CryoEM structures of the CHIK VLPs in complex with neutralizing antibody Fab fragments

Four cryoEM structures were determined to ~15 Å resolution of CHIK VLPs complexed with the Fab fragments of the neutralizing mouse MAbs CHK-9, CHK-152, m242 and m10 (*Figure 4*). All Fab fragments bound the VLPs with one Fab fragment per E2 molecule. Except for CHK-152, the concentration of the Fab molecules required for 50% neutralization was 100 or more times greater than the intact IgG (*Figure 5*). This implies a fundamental difference between how CHK-152 binds to

**Table 8.** Comparison between VEEV and CHIK VLPs.

| | E1-E2 | | | | TM | | | | Capsid | | | |
|---|---|---|---|---|---|---|---|---|---|---|---|---|
| | A1 | A2 | A3 | A4 | A1 | A2 | A3 | A4 | A1 | A2 | A3 | A4 |
| d in Å | 2.6 | 3.5 | 3.5 | 2.2 | 4.8 | 2.3 | 2.1 | 3.0 | 9.6 | 7.7 | 9.0 | 9.7 |
| κ in degrees | 2.3 | 3.6 | 3.7 | 3.4 | 5.4 | 8.2 | 10.4 | 10.0 | 9.7 | 8.0 | 10.0 | 11.7 |
| dCα in Å | 2.7 | 2.5 | 2.4 | 2.1 | 2.6 | 2.2 | 2.5 | 2.0 | 1.1 | 1.2 | 1.2 | 1.2 |
| num | 696 | 680 | 709 | 714 | 115 | 100 | 99 | 101 | 149 | 149 | 149 | 149 |
| % identity | E1 = 56%, E2 = 34% | | | | TM = 24% | | | | Capsid = 62% | | | |

Differences between the positions of the centers of mass (d) and orientation (κ) relative to the icosahedral symmetry axes of the (E1-E2) heterodimers, the trans membrane (TM) helices and capsid proteins at positions A1, A2, A3 and A4 (see *Figure 2*). The RMS distances (dCα) are given between the number (num) of equivalent Cα positions of the superimposed molecules A1 to A4. The percentage of identical amino acids in the aligned proteins is given on the last line of the table

the VLPs compared to the other Fabs. Because CHK-152 MAb and Fab fragment neutralize CHIKV at almost the same low concentration, it is unlikely that this MAb requires bivalent binding to inhibit infection.

The E1-E2 coordinates from the high-resolution VLP map were placed into the Fab bound cryoEM maps by aligning the icosahedral symmetry axes. The coordinates of the E2 B domain in the VLPs did not agree well with the B domain density in the maps of the VLP complexed with the CHK-9 or m242 Fab fragments, suggesting that the E2 B domain had a different orientation in these Fab-bound structures than in the non-complexed VLPs. To determine the modified position of the E2 B domain in these two complexes, the cryoEM density distributions were modified by setting to zero all density that was within 3 Å of all atoms in E1 and of all atoms in the A, C and D domains of E2. The crystallographic coordinates of the E2-B domain (*Voss et al., 2010*) were then fitted into the modified cryoEM density distributions, assuming T4 quasi-icosahedral symmetry.

The crystal structures of CHK-9 and m242 Fab fragments were determined to 3.0 Å and 3.1 Å resolution, respectively (*Table 9*). These Fab structures were fitted into the cryoEM maps of the corresponding VLP-Fab complexes. For the VLP-CHK-152 and VLP-m10 Fab complexes, for which no crystal structures of the corresponding Fab fragments had been determined, both the CHK-9 and m242 Fab crystal structures were tested to determine which Fab yielded the better fit to the cryoEM map of the complex. VLP-CHK-152 and VLP-m10 Fab cryoEM maps were best fitted with the Fab structures of CHK-9 and of m242, respectively (*Figure 6*). The T = 4 symmetry parameters, determined by fitting the E1-E2 heterodimer into the higher resolution VLP density, were used for fitting the Fab fragments into the cryoEM densities of the appropriate VLP-Fab complexes. The height of the density for the variable domains of the Fab fragments was comparable to that of the CHIKV glycoprotein, suggesting a nearly 100% binding occupancy of the Fab fragments under saturating binding conditions. The density for the constant domains (remote from the VLPs) of the Fab fragments was about 17% lower, consistent with the flexibility of the elbow angle between the constant and variable domains. The footprints of the Fab fragments on the surface of the VLPs were determined with the RIVEM program (*Xiao and Rossmann, 2007*) (*Figures 4,7*).

The footprint of CHK-9 and m242 Fabs onto the VLP localized primarily onto the domain A and did not include any part of domain B. Indeed, binding of these fragments to the VLPs had nudged domain B ~10 Å sideways, out of the way of the Fab fragment (*Figure 7*). In contrast, the footprint of the m10 Fab mapped exclusively on domain B and did not cause any observable movement (*Figure 7*). The footprint of CHK-152, however, spanned both domains B and A as well as the linker peptides to domain B represented by domain D. Thus, unlike the other Fabs, CHK-152 has the capacity to cross-link domain B with domain A. Alphaviruses become fusogenic at low pH when domain B, held in place by the flexible β-ribbons of domain D, becomes mobile and exposes the fusion loop on domain II of E1. As CHK-152 efficiently blocks viral fusion with host membranes (P Pal and M Diamond, unpublished observations), it may do so by cross-linking the flexible domain B to the rest of the virus surface. An analogous mechanism of inhibition was described for neutralization of West Nile virus, in which the viral envelope proteins were cross-linked by the bound FAB fragments (*Kaufmann et al., 2010*).

All four MAbs examined here bind to the end of the trimeric spikes that includes the putative receptor attachment region (*Davis et al., 1991*; *Dubuisson and Rice, 1993*; *Kinney et al., 1993*; *Klimstra et al., 1998*; *Bernard et al., 2000*; *Lee et al., 2002*; *Wang et al., 2003*; *Pierro et al., 2007*, *2008*; *Ryman et al., 2007*; *Li et al., 2010*), although only the CHK-9 Fab footprint actually overlaps the site (*Figure 7*). Nevertheless the proximity of these footprints to the possible receptor-binding site could probably allow neutralization by competitively limiting access of the virus to the cell surface receptor. To block formation of a fusogenic E1 trimer it is necessary to bind a Fab molecule to only one of the three E2 molecules in every spike. However, to block receptor attachment it is necessary to block all three receptor-binding sites on every spike. Such a stoichiometry would require substantially higher concentrations of Fabs to inhibit infectivity, as seen with CHK-9, m10 and m242 Fabs (*Figure 5*). The difference between MAb and Fab neutralization efficiency is presumably the result of bivalent attachment and cross-linking on a single virion or aggregation between virions. In summary our structural studies suggest that the CHK-9, m10, and m242 Fab fragments neutralize CHIKV infectivity by blocking the cellular binding site on the A domain of E2 and that the CHK152 inhibits fusion by stabilizing domain B of E2, preventing exposure of the fusion loop on E1.

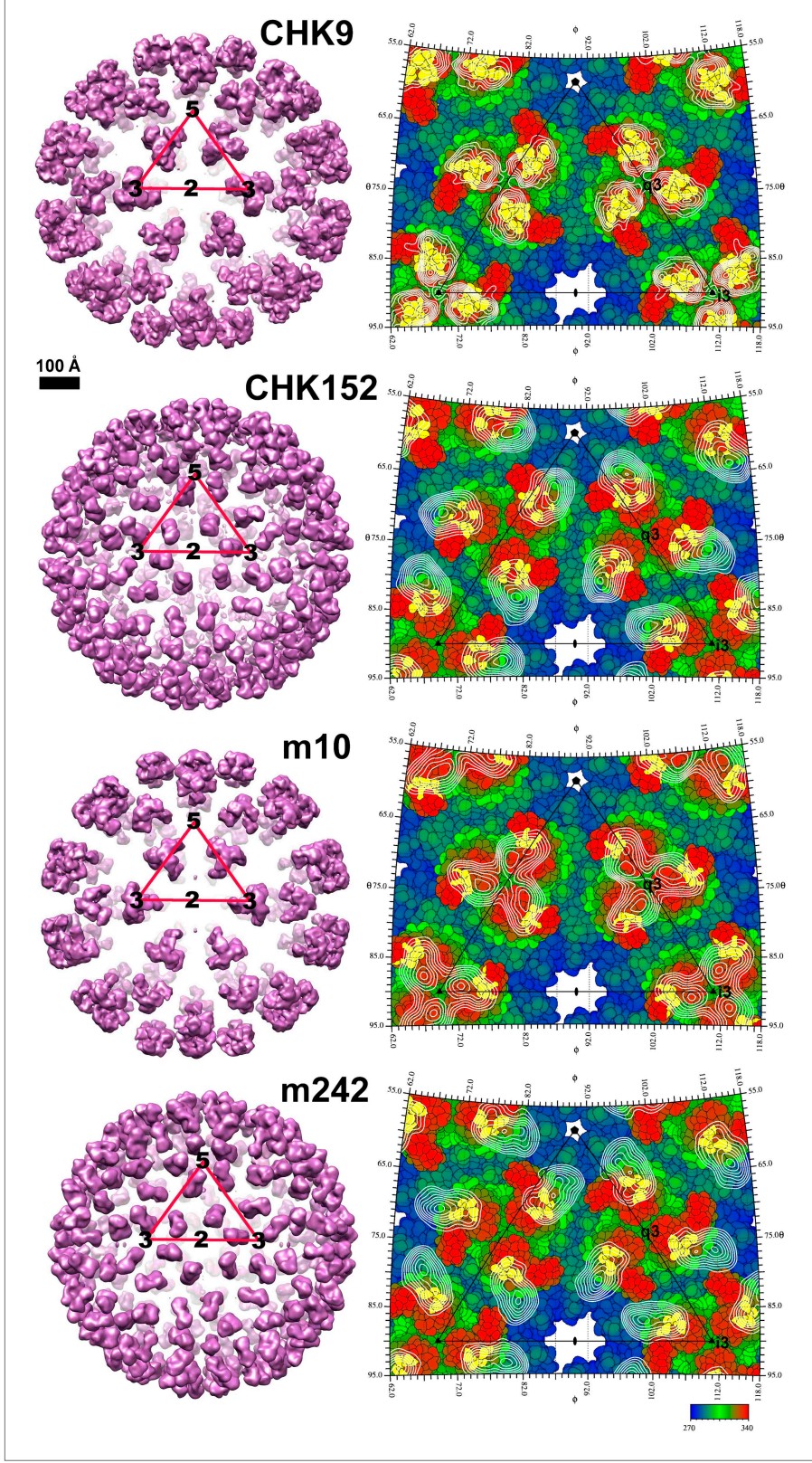

**Figure 4**. CHIKV VLP Fab complexes. Right: "Difference maps" showing the surface structure of the Fab molecules. The difference maps were produced by setting all density within 1.7 Å of an atom in the VLP structure to zero.
*Figure 4. Continued on next page*

*Figure 4. Continued*

Left: "Road maps" showing the projected surfaces of the VLP-Fab complexes for one (triangular) icosahedral asymmetric unit. The coloring is according to the radial distance of the surface from the center of the VLP. The footprints of the Fabs are shown in yellow. The radial projection of the whole Fab molecules onto the surface of the VLP is shown with white contours representing the height of the projected density. The mosaic background shows the amino acids that form the viral surface.

## Materials and methods

### CHIK VLP production and purification

CHIK VLPs have been reported as structural and immunologically indistinguishable from the mature infectious virus (*Akahata et al., 2010*); the use of CHIKV VLPs instead of live infectious virus has allowed us to avoid performing all structural studies under biosafety level 3 conditions. CHIK VLPs were produced and purified as described previously (*Akahata et al., 2010*). Following purification, the buffer was exchanged to PBS and the VLPs were concentrated to 1 mg/ml.

### Antibody production and purification of Fab fragments

MAbs m242, CHK-9, CHK-152, and m10 were purified from hybridoma superantants by protein A Sepahrose chromatography, and then digested with papain. The resultant Fab fragments were purified by sequential protein A Sepahrose and Superdex 75 16/60 size exclusion chromatography.

### Neutralization assays

Serial dilutions of MAbs or their Fab fragments were incubated with 100 focus-forming units (FFU) of CHIKV (CHIKV La Reunion 2006 OPY-1) for one hour at 37°C. MAb or Fab-virus complexes were added to Vero cells in 96-well plates. After 90 min, cells were overlaid with 1% (w/v) methylcellulose in Modified Eagle Media supplemented with 4% FBS. Plates were harvested 18 hr later, and fixed with 1% PFA in PBS. The plates were incubated sequentially with 500 ng/ml of mouse-human chimeric version of CHK-9 (containing a human γ1 constant region, P Pal and M Diamond, unpublished results) and horseradish peroxidase conjugated goat anti-human IgG in PBS supplemented with 0.1% saponin and 0.1% BSA. CHIKV-infected foci were visualized using TrueBlue peroxidase substrate (KPL), quantitated on an ImmunoSpot 5.0.37 macroanalyzer (Cellular Technologies Ltd), and analyzed using GraphPad Prism software.

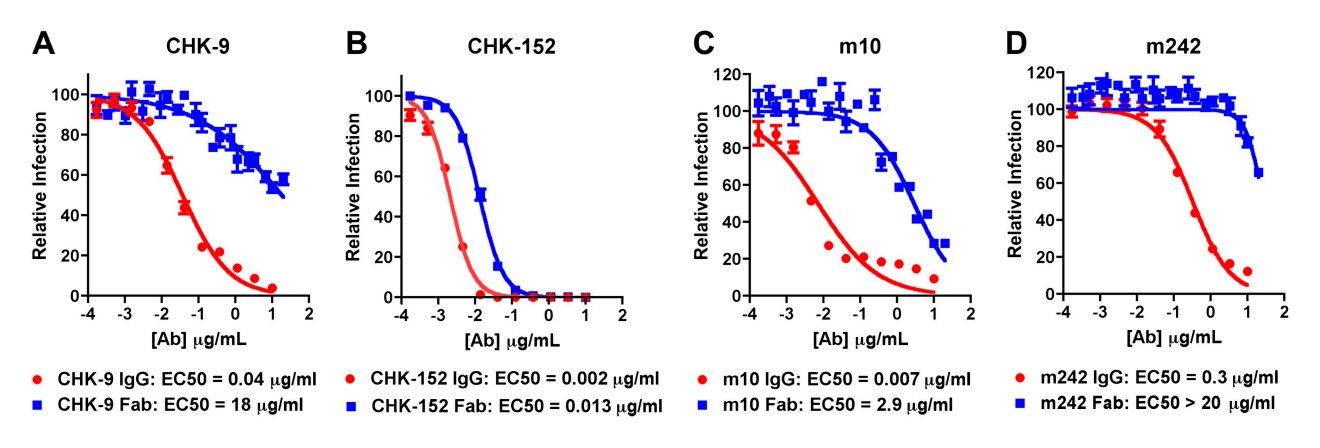

**Figure 5**. Inhibition of CHIKV by IgG and Fab fragments. Neutralizing activity of intact MAb and Fab fragments against CHIKV is shown for (**A**) CHK-9, (**B**) CHK-152, (**C**) m10 and (**D**) m242, as determined by the reduction in the number of focus-forming units (FFU) on Vero cells. MAbs and Fab fragments were mixed with 100 FFU of infectious CHIKV (strain La Reunion 2006 OPY-1) for one hour at 37°C before infecting Vero cells. Each data point is the average of three independent experiments performed in triplicate. Error bars represent standard deviations.

**Table 9.** Data collection and refinement statistics for the m242 and CHK-9 Fab molecules

|  | Fab m242 | Fab CHK9 |
|---|---|---|
| **Data collection** | | |
| X-ray source | 23-ID-B | 23-ID-B |
| Wavelength (Å) | 1.03 | 1.03 |
| Resolution (Å) | 3.1 | 3 |
| Space group | P2$_1$2$_1$2 | C2 |
| Unit cell (Å) | a = 137.0, b = 89.1, c = 94.4 | a = 87.9, b = 57.7, c = 118.1 |
| Unique reflections | 21,115 | 35,876 |
| Average redundancy | 3.7 | 5.8 |
| I/σ* | 17.5 (4.9) | 25.0 (5.0) |
| Completeness (%) | 98.3 (96.4) | 99.6 (99.9) |
| $R_{merge}$ (%)† | 10.1 (23.4) | 5.5 (39.2) |
| **Refinement** | | |
| Resolution (Å) | 3.1 | 1.8 |
| $R_{working}$ (%)‡ | 29.3 (34.5) | 18.7 (23.3) |
| $R_{free}$ (%)§ | 33.1 (34.3) | 22.0 (27.7) |
| rmsd bonds (Å) | 0.005 | 0.01 |
| rmsd angels (°) | 0.85 | 0.91 |
| # of residues Ramachandran disallowed | 0 | 1 |

*Values in parentheses throughout the table correspond to the outermost resolution shell

†$R_{merge}$ = Σ| I - <I> | / Σ I, where I is the measured intensity of reflections

‡$R_{working}$ = Σ||F$_{obs}$| -|F$_{calc}$|| / Σ|F$_{obs}$|

§$R_{free}$ has the same formula as R$_{working}$ except that calculation was made with the structure factors from the test set

## CryoEM imaging, processing and three-dimensional reconstruction of CHIK VLPs

Three microliter aliquots of purified CHIK VLP sample (1 mg/ml) were applied to 400 mesh C-flat grids (1.2 μm hole size) and double blotted inside a 100% humidified FEI vitrobot chamber, using a blotting time of 2 s. The grids were frozen by plunging into liquid ethane. A FEI Titan Krios electron microscope operated at 300 kV was used to collect cryoEM images on Kodak SO-163 films at a magnification of 59,000× and an electron dosage of ~25 e$^-$/Å$^2$ at the boiling point of liquid nitrogen. The microscope beam was aligned parallel with the optical axis of the microscope using the coma-free alignment technique (*Kimoto et al., 2003*). The quality of the alignment was checked by taking a CCD image of the carbon film under the same dosage as was used for data collection. The alignment was accepted when the Thon rings on the carbon support were visible beyond (1/4.5) Å$^{-1}$. A total of 1532 films were developed using full strength D19 (Kodak) solution. Micrographs were digitized with a Nikon Coolscan 9000ED scanner at a step size of 6.35 μm/pixel, which yielded a pixel size of 1.05 Å/pixel on the specimen. The absolute pixel size was determined by varying the pixel size from 1.05 to 1.14 in steps of 0.01 Å to find the "best" fit of the CHIKV E1E2 heterodimer crystal structure into the final cryoEM density. The fit was based on the average density of all the non-hydrogen fitted atoms (sumf). The highest value of sumf was found when the pixel separation was 1.11 Å.

Of the 1532 micrographs, 1012 did not have drift or charging problems and had Thon rings beyond (1/6.0) Å$^{-1}$ observed through the sample. These were selected for further processing. A total of 52,183 particles were boxed with the EMAN2 program e2boxer (*Tang et al., 2007*). The selected micrographs were found to be under-focused by between 1.1 μm and 2.9 μm using the EMAN program CTFIT (*Ludtke et al., 1999*). Initially a "four-fold binned" map was used in which each set of 4 × 4 pixels was replaced by one pixel with the average height of the original sixteen pixels. Thus, the pixels in this map were separated by 4.44 Å, limiting the resolution to ~13 Å. An initial CHIK VLP model was produced by selecting particles with 5- 3- and 2-fold symmetric projections from which an icosahedral reconstruction was made. This model was iteratively refined using a coarse 5° angular step size to improve the orientation and positioning of each particle until convergence was reached, as indicated by the lack of change of the Fourier shell correlation between successive cycles. At this point, the model had the characteristics of an alphavirus (*Cheng et al., 1995*). In further iterations, the model was modified progressively by setting to zero both the inner density corresponding to the RNA genome and also the outer noise background density. Orientation and centers for each of the particles were determined by comparing the observed two-dimensional projections with projections of the model generated using a 1° angular step size. Each observed projection was allocated to the three best fits in the calculated projection classes. The images associated with each class were averaged and used to generate an improved model for the next cycle. Only those particles whose orientations and centers had changed their orientation and position by less than 2° and 2 pixels

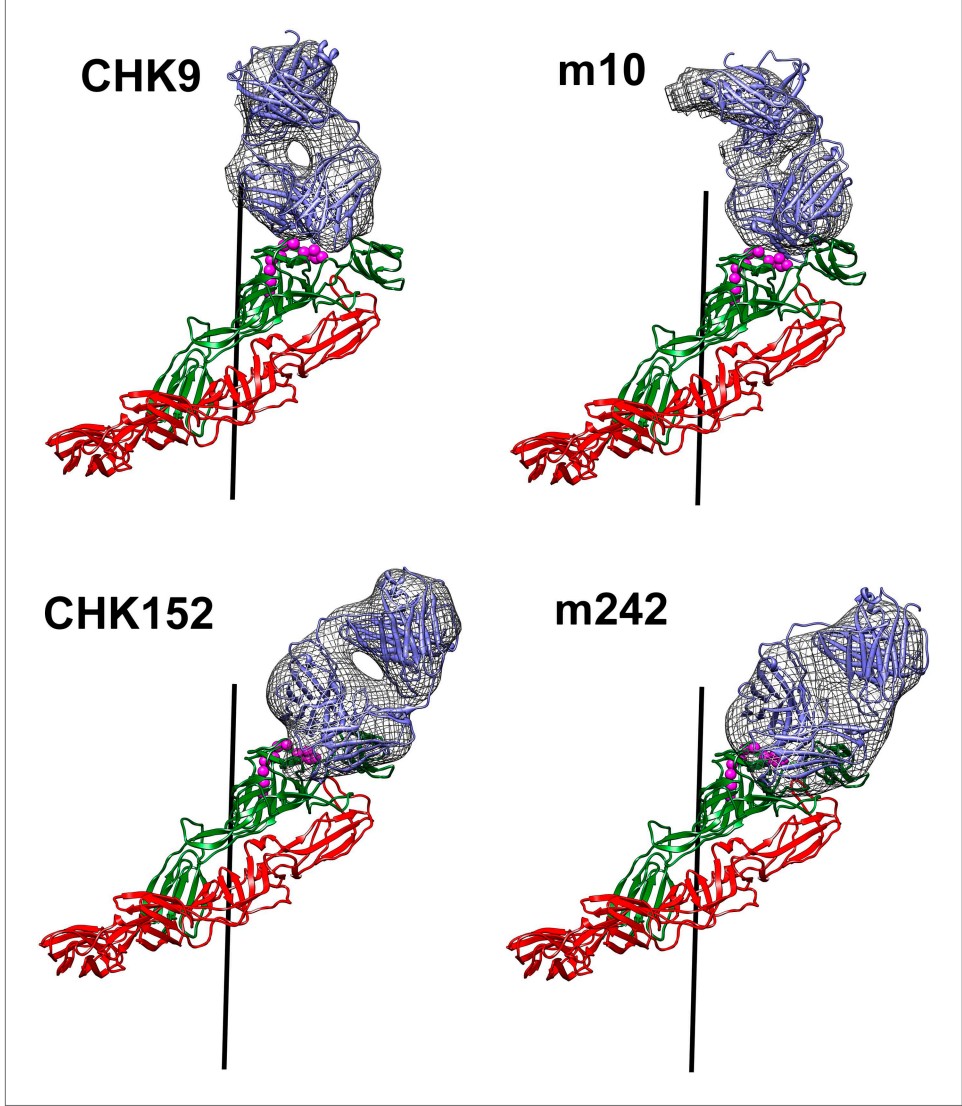

**Figure 6**. Fit of the Fab structures (blue) into the cryoEM density "difference" maps (hatched grey surface) calculated as described for *Figure 4*. Shown also is how the Fab molecules bind to the E1(red)-E2 (green) heterodimer.

between successive cycles were retained for the next reconstruction. Unstable particles were discarded after each cycle and were not reconsidered.

When the global searches had converged, local fine angular and positional refinements were conducted using 2-fold instead of 4-fold binned images. Particles were discarded when their three top solutions did not converge. After each cycle, the resolution of the map was estimated and used to modify the model by filtering out the Fourier coefficients beyond the current resolution. Further improvement was hindered because it was assumed that the magnification of each micrograph was the same and that the defocus distance of each particle on a given micrograph was the same. The program Frealign (*Grigorieff, 2007*) was used to refine orientation, position, defocus distance and relative pixel size for each micrograph using the original scanned data without binning. Particles were rejected that were found after refinement to have changes in any of these parameters by more than 2σ. Overall convergence was established when the plot of the Fourier shell correlation vs resolution did not improve as estimated by visual inspection. The final map was reconstructed using 36,236 particles. The resolution of this map was estimated to be 4.6 Å, based on the spacing at which the Fourier shell correlation fell below 0.5 (*Van Heel, 1987*). To verify the effective resolution of

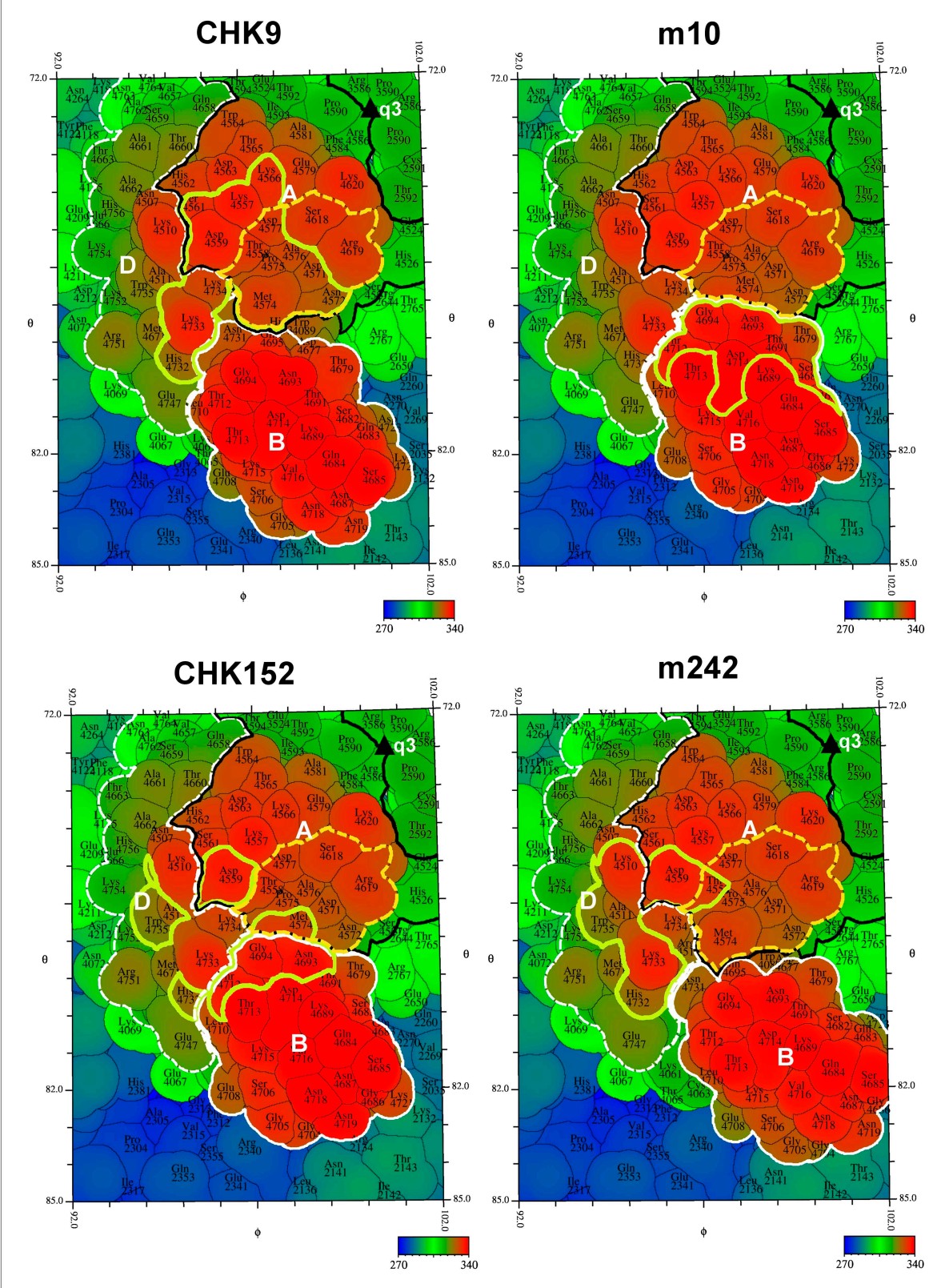

**Figure 7**. "Road maps" showing footprint of four neutralizing Fabs on the VLP surface at position #3 as defined in **Figure 2**. In order to differentiate between amino acids in different quasi 3-fold related subunits, their identity is defined as the amino acid sequence number in E1 + 2000, 3000, and

*Figure 7. Continued on next page*

*Figure 7. Continued*

4000, and in E2 + 2500, 3500, and 4500 for positions #2, 3, and 4, respectively (see *Figure 2*). The surface is colored according to radial distance from the center of the VLP. The A, B, and D domains of E2 are bounded by a black, white and dashed white line, respectively. Residues in the putative receptor binding site on domain A of E2 are bounded by a yellow dashed line. The footprint of the Fabs onto the VLP surface is outlined in yellow.

the map (*Chang et al., 2012*), the orientation of each image was determined with respect to the final map after having been low passed filtered to 7 Å. The resultant orientations of the images were used to calculate the Fourier shell correlation between two equal random groups of all the images. The map resolution was found to be 5.3 Å using the 0.5 Fourier shell correlation criterion. A series of maps were calculated using a B factor applied to the Fourier coefficients of the inverted map to estimate the noise level and backbone density continuity. The best value of B was -40 Å$^2$, which slightly increased the emphasis on the high order terms.

## Fitting of CHIKV crystal structure into the EM map

Initially, the E1-E2 heterodimer crystal structure was fitted as a rigid body close to an icosahedral 3-fold axis (*Figure 2*) defined by the polar angles psi = 69.1°, fi = -90° (*Rossmann and Blow, 1962*). This position was called "#1" (*Figure 2A*). The icosahedral 3-fold axis was used to generate the complete icosahedral "i3" spike. Next, the i3 spike was rotated by the quasi-2-fold axis at psi = 74.6°, fi = -80.9° (an orientation appropriate for T = 4 quasi-symmetry) to produce the "q3" spike, generating positions #2, #3, #4 of the heterodimer (*Figure 2A*). The position of this quasi-2-fold axis was refined to maximize the average height of the fitted density (sumf) while minimizing the number of clashes between symmetry related E1-E2 heterodimers (*Rossmann et al., 2001*).

The resultant positions of the four quasi-equivalent CHIKV heterodimers were used as starting positions for further position and orientation refinement of individual domains. E1 was divided into domain I (residues 1-36, 132-168, 273-293), II (residues 37-131, 169-272) and III (residues 294-393). E2 was divided into domain A (residues 16-134), B (residues 173-231), C (residues 269-342), and D the β-ribbon connector (residues 7-15, 135-172, 232-268). These domains were fitted sequentially (I, II, III, A, C, B, and D). After each domain was fitted independently, the map was modified by zeroing out the density around each fitted atom within a radius of 3.0 Å. Thus, when fitting the next domain, atoms placed into zero density acted as a restraint by reducing the overall fitting criterion, "rcrit" (*Rossmann et al., 2001*). This process substituted for avoiding clashes had all domains been fitted simultaneously. After the individual domains had been fitted, the bond geometry between the carboxy end of one domain and the amino end of the next domain were regularized.

The average density height of the final atomic structure (sumf) was better for the individual domain fitting results than for the original T = 4 rigid body fitting results (*Table 1A*), validating the fitting strategy. The quality of the independently fitted domain structure was also validated by comparing the fit of the atomic structures at each of the four quasi-equivalent densities. The density of each non-hydrogen atom in the CHIKV model was determined by interpolation using the densities at the eight surrounding grid points. The average density ($\rho_i$) of residue i was then set to the average densities of all the atoms associated with this residue. The correlation coefficient (CC) was calculated between the densities $\rho_i(A)$ and $\rho_i(B)$ where these are the cryoEM densities of residue i at the quasi-equivalent positions A and B. The sums are taken over i = 1, N where N is the number of residues being matched.

$$CC = \frac{\sum[(\rho_i(A) - <\rho_i(A)>)(\rho_i(B) - <\rho_i(B)>)]}{[\sum(\rho_i(A) - <\rho_i(A)>)^2 \; \sum(\rho_i(B) - <\rho_i(B)>)^2]^{1/2}}$$

## Identifying non-covalent contacts between molecules that form the CHIKV VLPs

Non-covalent contacts between molecular entities were determined and classified by identifying the number of times a specific atom in one molecule was within a defined distance, Dlim = 3.5 Å, of any atom in a specific neighboring molecule. The contacts of each atom in a given molecule were added

to yield a measure of how many close contacts the first molecule had with the second molecule. This count is roughly equivalent to measuring the surface contact area between the two molecules. In addition to measuring the total number of contacts, a count was made of that portion of contacts that represented hydrophobic interactions by counting only carbon-carbon atom distances less than Dlim. Also, the number of potential hydrogen bonds was estimated by counting the number of short (Dlim < 3.5 Å) distances between oxygen and nitrogen atoms. Lastly, a count was made of possible formation of salt bridges by counting the number of occasions that an Arg, Lys or His residue approached to within Dlim of an Asp or Glu residue.

### Radius of gyration of the trimeric spikes

The radius of gyration of the spikes was computed as:

$$r_g = \sqrt{\frac{\sum (r_i^2)}{N}},$$

where the Sum (i = 1 to N) is taken over the N Cα atoms in E1 and E2 of one spike, and $r_i$ is the radius of the ith atom measured from the spike axis. The latter is defined by the positions of the centers between the 3-fold (in the i3 spike), or quasi-3-fold (in the q3 spike) related atoms.

### Comparison of VEEV and CHIK VLPs

Alignment of the molecular components between VEEV and CHIK VLPs was performed with the HOMOlogy program (*Rossmann and Argos, 1975*). The position and orientation of each molecular component was calculated with respect to the equivalenced Cα atoms.

### CryoEM imaging, processing and three-dimensional reconstruction of the Fab-CHIK VLP complexes

CHIK VLP particles were incubated with Fab fragments at 4°C for two hours using a stoichiometric ratio of about four Fab fragments per E2 molecule. Samples were hand-blotted, flash-frozen on holey grids in liquid ethane. Images were recorded at 47,000× magnification with a CM300 field emission gun microscope using electron dose levels of approximately 20 electrons per Å². All micrographs were digitized at 6.35 μm per pixel using a Nikon scanner. Individual particle images were boxed using the program e2boxer in the EMAN2 package. Subsequently, the boxed images were two-fold binned, resulting in a sampling of the specimen images at 2.69 Å intervals. The CTFIT program from the EMAN package was used to determine the contrast transfer function parameters. A cryoEM reconstruction of CHIK VLP, low pass filtered to 18 Å, was used as an initial model for orientation determination and further refinement. The number of particles incorporated into the final reconstruction was 1728, 1820, and 1599 for Fabs m242, CHK9 and m10, giving final resolutions of 15.6 Å, 14.9 Å and 16.9 Å on the basis of a 0.5 Fourier shell correlation threshold, respectively.

### Crystallization and structure determination of the m242 and CHK-9 Fab fragments

The m242 Fab was crystallized in 2M ammonium sulfate and 0.1 M sodium acetate pH 4.5 and the CHK-9 Fab was crystallized in 25% PEG 3350, 0.1M Tris-Cl pH 8.5 and 0.2M lithium sulfate. Crystals were grown by vapor diffusion in hanging drops at 20°C. Crystals were flash-frozen and data were collected at 100K at the Advanced Photon Source (APS) GM/CA-CAT 23-ID-B beamline. Data were processed with the HKL2000 program (*Otwinowski and Minor, 1997*). The m242 Fab crystals diffracted to 3.1 Å resolution and had a P2$_1$2$_1$2 space group with cell dimensions a = 137.0, b = 89.1 and c = 94.4 Å. There were two Fab molecules per asymmetric unit. The CHK-9 crystals diffracted to 3.0 Å resolution and had a C2 space group with cell dimensions a = 87.9, b= 57.7 and c = 118.1 Å. There was one Fab molecule per asymmetric unit. The structures of the m242 Fab and CHK-9 Fab crystals were determined by molecular replacement using a Fab crystal structure (Protein Data Bank: 3DGG) as a search model with the program Phaser (*Mccoy et al., 2007*). The structures were refined using the program Refmac (*Murshudov et al., 1997*) (*Table 9*). The rms difference between the Cα atoms of the two molecules in the asymmetric unit of the m242 Fab crystals was 0.7 Å.

## Acknowledgements

We thank the referees for pointing out that "a lesson to be drawn from this work is the merit of obtaining a cryoEM structure even at a lower resolution than the crystal structure to reveal the structure of the virus proteins free of potential crystallization artifacts and close to the solution state conformation".

We thank Syd Johnson (MacroGenics) for the chimeric CHK-9 Mab, the sequence of the CHK-9 Fab fragment, and purifying CHK-9 and CHK-152 IgG. We are grateful to Josh Yoder for preparing the m242 Fab fragment. Use of the Advanced Photon Source was supported by the U.S. Department of Energy, Office of Science, Office of Basic Energy Sciences, under Contract No. DE-AC02-06CH11357. We thank Sheryl Kelly for formatting the manuscript.

## Additional information

### Funding

| Funder | Grant reference number | Author |
| --- | --- | --- |
| National Institutes of Health | AI095366 | Siyang Sun, Ye Xiang, Heather Holdaway, Xinzheng Zhang, Michael G Rossmann |
| National Institutes of Health | AI104545 | Pankaj Pal, Michael S Diamond |

The funders had no role in study design, data collection and interpretation, or the decision to submit the work for publication.

### Author contributions

SS, Collected cryoEM data and produced cryoEM reconstructions, Determined the crystal structures of the Fab fragments and performed the model fitting, Wrote and critiqued the paper; YX, Collected cryoEM data and produced cryoEM reconstructions, Wrote and critiqued the paper; WA, Produced the CHIK VLPs as well as the m10 and m242 MAbs and Fab fragments; HH, Helped with cryoEM data collection; PP, Produced the CHK-9 and CHK-152 MAbs and Fab fragments, and performed all neutralization assays; XZ, Re-determined the resolution of the CHIK VLP cryoEM map and helped in making additional figures suggested by the reviewers; MSD, Produced the CHK-9 and CHK-152 MAbs and Fab fragments and performed all neutralization assays, Wrote and critiqued the paper; GJN, Produced the CHIK VLPs as well as the m10 and m242 MAbs and Fab fragments, Wrote and critiqued the paper; MGR, Analyzed the fitted structures, wrote and critiqued the paper

## Additional files

### Major datasets

The following datasets were generated:

| Author(s) | Year | Dataset title | Dataset ID and/or URL | Database, license, and accessibility information |
| --- | --- | --- | --- | --- |
| Sun S, Xiang Y, Rossmann MG | 2012 | Chikungunya virus neutralizing antibody m242 Fab fragment | PDB ID 4GQ8 | Available at www.PDB.org. |
| Sun S, Xiang Y, Rossmann MG | 2012 | Chikungunya virus neutralizing antibody CHK9 Fab fragment | PDB ID 4GQ9 | Available at www.PDB.org. |
| Sun S, Xiang Y, Rossmann MG | 2012 | Chikungunya virus VLP pseudo-atomic resolution coordinates | PDB ID 312W | Available at www.PDB.org. |
| Sun S, Xiang Y, Rossmann MG | 2012 | Chikungunya virus 5.3 Ångström resolution cryoEM map | EMD-5577 | Available at www.ebi.ac. uk/pdbe/emdb/. |
| Sun S, Xiang Y, Rossmann MG | 2012 | Chikungunya virus VLP complexed with Fab CHK9 (Fab co-ordinates) | PDB ID 3J2Y | Available at www.PDB.org. |

| Sun S, Xiang Y, Rossmann MG | 2012 | Chikungunya virus 15 Ångström resolution cryoEM map complexed with Fab CHK9 | EMD-5578 | Available at www.ebi.ac.uk/pdbe/emdb/. |
|---|---|---|---|---|
| Sun S, Xiang Y, Rossmann MG | 2012 | Chikungunya virus VLP complexed with Fab m242 (Fab co-ordinates) | PDB ID 3J2X | Available at www.PDB.org. |
| Sun S, Xiang Y, Rossmann MG | 2012 | Chikungunya virus complexed with Fab m242, 15 Ångström resolution cryoEM map | EMD-5576 | Available at www.ebi.ac.uk/pdbe/emdb/. |
| Sun S, Xiang Y, Rossmann MG | 2012 | Chikungunya virus complexed with Fab CHK152, interpreted with Fab CHK9 (Fab co-ordinates) | PDB ID 3J30 | Available at www.PDB.org. |
| Sun S, Xiang Y, Rossmann MG | 2012 | Chikungunya virus VLP complexed with Fab CHK152, interpreted with Fab CHK9, 15 Ångström resolution cryoEM map | EMD-5580 | Available at www.ebi.ac.uk/pdbe/emdb/. |
| Sun S, Xiang Y, Rossmann MG | 2012 | Chikungunya virus complexed with Fab m10, interpreted with Fab m242 (Fab coordinates) | PDB ID 3J2Z | Available at www.PDB.org. |
| Sun S, Xiang Y, Rossmann MG | 2012 | Chikungunya virus complexed with Fab m10, 15 Ångström resolution cryoEM map | EMD-5579 | Available at www.ebi.ac.uk/pdbe/emdb/. |

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
