## [Decision Letter]

Thank you for choosing to send your work entitled “Structural analyses at pseudo atomic resolution of Chikungunya virus antibody neutralization mechanisms” for consideration at *eLife*. Your article has been favorably evaluated by a Senior editor and 2 reviewers, one of whom is a member of our Board of Reviewing Editors. The following individuals responsible for the peer review of your submission want to reveal their identity: Werner Kühlbrandt (Reviewing editor).

The Reviewing editor and reviewer discussed their comments before we reached this decision, and the Reviewing editor has assembled the following comments to help you prepare a revised submission.

The manuscript is a major advance on the cryo-EM structure of the same virus at considerably lower resolution (18 Å) that was published by the same lab in Nature Medicine in 2010. The present manuscript offers new insights at much higher resolution (around 5 Å, see below) not only into the virus capsid structure itself, but also (although at considerably lower resolution) into how the capsid interacts with neutralizing antibodies. This is potentially of medical interest, given that there is at present no approved vaccine against the virus, which is a major human pathogen.

A technical highlight in this paper is the analysis of the docking of the E1 and E2 crystal structures into the cryo-EM maps of individual domains of these two glycoproteins in CHIKV particle. This led to the finding that the cryo-EM models of the relative positions of the domains of these proteins are consistent among the 4 quasi-equivalent heterodimers within the asymmetric unit of the icosaheral particle, but different from those of the crystal structures. Consequently, the molecular contacts between the subunits are more extensive in the cryo-EM structure than in the crystal structure. The authors attribute these differences to the crystal packing. A lesson to be drawn from this work is the merit of obtaining a cryo-EM structure even at a lower resolution than the crystal structure to reveal the structure of the virus proteins free of potential crystallization artifacts and close to the solution state conformation.

Nevertheless, the reviewers had some concerns that need to be addressed in a revised manuscript:

1. Since this is not the first high-resolution cryo-EM map of a virus in this family (see Zhang et al, EMBO J, 30, 3854-3863 (2011)), the authors should focus more on the structural differences between the two viruses and on discussing biomedical implications.

2. The map resolution (4.6 Å) is likely to be an overestimate. At 4.6 Å, beta strands should be easily resolved throughout the map. The resolution should be assessed by procedures that reduce the risk of over-fitting noise, such as those recently described by Scheres et al (Nature Methods 9, 383-396, 2012), or by refining the map against data in which phases above a certain resolution (e.g., 7 Å) have been randomized. These tests are increasingly accepted as the “gold standard” in single-particle cryo-EM. It is anticipated that the actual resolution will then turn out to be no better than 5 Å, but this will not distract from the quality of the map or the interest and importance of the work, but it will help to maintain rigorous resolution standards in the cryo-EM field.

3. A few figures that show the quality of the fit should be added, to provide a sense of the match between model and density throughout, including the alpha carbon backbone fit to the density for the trans-membrane region. We presume that this information will also be retrievable from EMDB and PDB when the paper is published.

4. The locations of individual subunits in an asymmetric unit have been shown in many previous studies. Figure 2 can 3 can be combined to show the number of subunits per asymmetric units and their inter-subunit interactions.

---

## [Author Response]

*1. Since this is not the first high-resolution cryo-EM map of a virus in this family (see Zhang et al, EMBO J, 30, 3854-3863 (2011)), the authors should focus more on the structural differences between the two viruses and on discussing biomedical implications*.

This is clearly a topic we should have covered in our paper. We have now made a systematic comparison of the VEEV structure (Zhang et al 2011) with the CHIKV VLP structure reported by us in this paper. We have added a new section “Comparison with the VEEV structure” just prior to the section on the antibody complexes. The results are interesting, as explained in this new section.

*2. The map resolution (4.6 Å) is likely to be an overestimate. At 4.6 Å, beta strands should be easily resolved throughout the map. The resolution should be assessed by procedures that reduce the risk of over-fitting noise, such as those recently described by Scheres et al (Nature Methods 9, 383-396, 2012), or by refining the map against data in which phases above a certain resolution (e.g., 7 Å) have been randomized. These tests are increasingly accepted as the “gold standard” in single-particle cryo-EM. It is anticipated that the actual resolution will then turn out to be no better than 5 Å, but this will not distract from the quality of the map or the interest and importance of the work, but it will help to maintain rigorous resolution standards in the cryo-EM field*.

To use the first method suggested by the referee would mean a completely new start to the structure determination. This would require months of dedicated work and we suggest is not feasible at this stage of the CHIK VLP analysis. However, we have pursued the referees' second suggestion. We re-determined the orientation of each boxed image against a 7 Å low-pass filtered map of our current model. These new orientations were then used for a Fourier shell coefficient analysis in which the images had been divided into two equal groups. The result gave a resolution of 5.3 Å using a Fourier correlation of 0.5 as a cut off. We have added these results to the manuscript and we have corrected the 4.6 Å claim to 5.3 Å in all relevant places.

*3. A few figures that show the quality of the fit should be added, to provide a sense of the match between model and density throughout, including the alpha carbon backbone fit to the density for the trans-membrane region. We presume that this information will also be retrievable from EMDB and PDB when the paper is published*.

We have added the following figures to show the quality of the structure fitted to the cryo-EM density: Figure 3A, the E1-E2 heterodimer; Figure 3B, the trans-membrane helices. In each case we have chosen the quasi-equivalent structure closest to the icosahedral 5-fold axis.

*4. The locations of individual subunits in an asymmetric unit have been shown in many previous studies. Figure 2* can *3* can *be combined to show the number of subunits per asymmetric units and their inter-subunit interactions*.

We have combined these figures as suggested.